

# Modeling atmospheric brown carbon in the GISS ModelE Earth system model

Maegan A. DeLessio[1,2], Kostas Tsigaridis[3,2], Susanne E. Bauer[2,3], Jacek Chowdhary[4,2], Gregory L. Schuster[5]

[1]Department of Earth and Environmental Sciences, Columbia University, New York, NY 10025, USA
[2]NASA Goddard Institute for Space Studies, New York, NY 10025, USA
[3]Center for Climate Systems Research, Columbia University, New York, NY 10025, USA
[4]Department of Applied Physics and Applied Mathematics, Columbia University, New York, NY 10025, USA
[5]NASA Langley Research Center, Hampton, VA 23661, USA

*Correspondence to*: Kostas Tsigaridis (kostas.tsigaridis@columbia.edu)

**Abstract.** Brown carbon (BrC) is an absorbing organic aerosol, primarily emitted through biomass burning, that exhibits light absorption unique from both black carbon (BC) and other organic aerosols (OA). Despite many field and laboratory studies seeking to constrain BrC properties, the radiative forcing of BrC is still highly uncertain. To better understand it's

climate impact, we introduced BrC to the One-Moment Aerosol (OMA) module of the GISS ModelE Earth system model (ESM). We assessed ModelE sensitivity to primary BrC processed through a novel chemical aging scheme, as well as secondary BrC formed from biogenic volatile organic compounds (BVOCs). Initial results show BrC typically contributes a top of the atmosphere (TOA) radiative effect of 0.04 W m⁻². Sensitivity tests indicate that explicitly simulating BrC (separating it from other OA), including secondary BrC, and simulating chemical bleaching of BrC all contribute

distinguishable radiative effects and should be accounted for in BrC schemes. This addition of prognostic BrC to ModelE allows for greater physical and chemical complexity in OA representation with no apparent trade-off in model performance as evaluation of ModelE aerosol optical depth, with and without the BrC scheme, against AERONET and MODIS retrieval data reveals similar skill in both cases. Thus, BrC should be explicitly simulated to allow for more physically based chemical composition, which is crucial for more detailed OA study like comparisons to in-situ measurement campaigns. We include

additional recommendations for BrC representation within ESMs at the end of this paper.

## 1 Introduction

Carbonaceous aerosols are important, short-lived climate forcers. Black carbon (BC), a strongly absorbing carbonaceous aerosol produced from fuel and biomass burning (BB), contributes a significant positive radiative forcing (RF) to the atmosphere (Hansen et al., 1997; Jacobson, 2001; Ramanathan and Carmichael, 2008; Bond et al., 2013). The Sixth

Assessment Report (AR6) of the Intergovernmental Panel on Climate Change (Calvin et al., 2023) estimates a BC effective RF of 0.11 W m⁻² (Szopa et al., 2021). BB also emits organic aerosols (OA) (Ito and Penner, 2005), another type of



carbonaceous aerosol, which cool the atmosphere at an estimated RF of -0.21 W m$^{-2}$ (Szopa et al., 2021). Beyond BB, secondary production is a key source of OA: isoprene and other biogenic or anthropogenic volatile organic compounds (VOCs) partition and react in the atmosphere to form secondary organic aerosols (SOA; Shrivastava et al., 2017; Mahilang et al., 2021).

As warming temperatures and changes in precipitation drive increases in wildfire frequency and intensity (Flannigan et al., 2009; Keywood et al., 2013), and cleaner technologies possibly lead to a further reduction of other aerosol sources (Bauer et al., 2022), carbonaceous aerosols including OA could possibly become more prominent. Observations at Whiteface Mountain, downwind of the U.S. West coast BB region, have already shown an increase in OA found in cloud water over the last ten years, suggesting a growing influence of wildfire smoke (Lawrence et al., 2023). SOA from biogenic VOCs (BVOCs) are also expected to grow in importance; SOA burden could possibly be greater than that of sulfate aerosols by 2100 (Tsigaridis and Kanakidou, 2007). Despite their growing importance, OA still pose a large gap in aerosol modeling: the IPCC estimated an OA RF uncertainty of 0.23 W m$^{-2}$, about the same magnitude as the cooling effect itself (Szopa et al., 2021). Improving the physical and chemical parameterization of OA in climate models can allow for better calculation of OA forcing. To make such an improvement, light absorption of brown carbon aerosols must be accounted for.

Brown carbon (BrC) refers to the subset of OA that absorb light (Andreae and Gelencsér, 2006). Since the chemical composition, and therefore absorptivity, of these aerosols vary greatly, BrC can be best thought of as a classification of aerosols, rather than a specific compound or compounds class. Typically, BrC contains absorbing organic chromophores such as nitroaromatics, polyaromatic hydrocarbons (PAHs), or lignin-derived compounds (Samburova et al., 2016; Lin et al., 2018; Fleming et al., 2020). It is emitted by incomplete combustion and smoldering fires (McMeeking et al., 2009; Chakrabarty et al., 2010). Though its main source is BB, secondary BrC can form through gaseous and aqueous reactions as SOA (Lee et al., 2014).

BrC has a spectrally-dependent absorption in the UV-to-visible wavelength range, strongly absorbing in the UV/near-UV but much less in the rest of the visible spectrum, hence its color and name (Andreae and Gelencsér, 2006; Laskin et al., 2015). It is this absorption pattern that distinguishes BrC from BC, which is co-emitted by fires (Lack et al., 2012; Saleh et al., 2014; Pokhrel et al., 2016), as BC is a stronger absorber across all visible wavelengths and into the near-IR (Bond and Bergstrom, 2006). There are recent laboratory and field studies that have observed "dark BrC", suggesting a distinct class of refractory, highly absorbing BrC (Hoffer et al., 2017; Saleh et al., 2018; Corbin et al., 2019; Chakrabarty et al., 2023), also co-emitted with BC, that can be best described as resembling tar balls (Pósfai et al., 2004; Alexander et al., 2008). Because there is limited characterization of these aerosols, and initial work by Chakrabarty et al. (2023) suggests its single-scattering albedo (SSA) and absorption Ångström exponent (AAE) are indistinguishable from that of BC, we did not explicitly represent this subset of BrC.

Like most aerosols, BrC undergoes processing, or aging, in the atmosphere. Heterogenous oxidation by hydroxyl (OH) and nitrate (NO$_3$) radicals can lead to functionalization of BrC compounds (Cheng et al., 2020; G. Schnitzler et al., 2020), while aqueous oxidation can form oligomeric BrC (Hems et al., 2020). These processes cause an increase in absorption,



known as browning. Further oxidation by OH, photolysis, or ozonolysis result in fragmentation of BrC compounds and subsequent decreases in absorption, known as bleaching (Hems et al., 2021). Laboratory studies have shown that primary BrC undergoes browning followed by bleaching, while secondary BrC only bleaches (Zhao et al., 2015). Other properties of BrC can also change with chemical aging. Volatility typically decreases with functionalization (browning) and increases with fragmentation (bleaching). As a direct result of this chemical processing, molecular weight typically increases with browning and decreases with bleaching (Di Lorenzo and Young, 2016; Di Lorenzo et al., 2017).

Most literature on BrC properties, such as composition, absorption, size, and atmospheric processing, has come from laboratory studies of BrC proxies or lab burns (Saleh et al., 2014; Di Lorenzo and Young, 2016; Liu et al., 2016; Tang et al., 2016; Lin et al., 2018; Al Nimer et al., 2019; Shetty et al., 2019; Wong et al., 2019; Li et al., 2020a). In-situ BrC absorption, mass, and size distribution have been measured during flight campaigns like DC3 and SEAC$^4$RS (Zhang et al., 2017), ATom (Zeng et al., 2020), WE-CAN (Zeng et al., 2021), and FIREX-AQ (Washenfelder et al., 2022; Zeng et al., 2022), in or downwind of fires. There have also been several studies that have retrieved BrC aerosol properties from observations outside of laboratories and flight campaigns. These studies utilized retrieval data from AERONET (Arola et al., 2011, 2015; Schuster et al., 2016) or IMPROVE (Chow et al., 2018) ground-based networks, as well as satellite retrievals (Li et al., 2020b, 2022), relying on the differences in optical properties, or optical properties and size, between BrC and other absorbing aerosols, like BC and dust.

There have been several studies that have modelled BrC in chemical transport models, all of which either use GEOS-Chem (Park et al., 2010; Wang et al., 2014; Saleh et al., 2015; Jo et al., 2016; Wang et al., 2018; Tuccella et al., 2020; Carter et al., 2021; Zhu et al., 2021) or IMPACT (Feng et al., 2013; Lin et al., 2014). Only three studies have shown implementations of BrC in Earth system/climate models (Brown et al., 2018; Zhang et al., 2020; Drugé et al., 2022). Both Brown et al. (2018) and Zhang et al. (2020) simulated BrC using the Community Atmosphere Model within the Community Earth System Model (CESM), while Drugé et al. (2022) used the Centre National de Recherches Météorologiques (CNRM) climate model. Zhang and Drugé separately simulated BrC from other OAs. Zhang treated a prescribed portion of OA as brown, and Drugé assumed BB OA is brown and fossil fuel OA is non-absorbing. In contrast, Brown considered BrC and OA as all one species. All three studies included a bleaching parameterization for BrC, though none included a browning parameterization.

In this study, we present the first implementation of BrC aerosols in the GISS ModelE Earth system model (Kelley et al., 2020; Bauer et al., 2020). We introduced BrC into the One-Moment Aerosol (OMA) module of ModelE by defining four key properties or processes: BB emissions of primary BrC, formation of secondary BrC, optical properties of BrC tracers, and chemical aging of primary BrC. This constitutes an improvement in simulating OA absorption in ModelE, as BrC was previously not explicitly represented, and all OA were assumed to be slightly absorbing (Koch, 2001), consistent with other treatments of OA (Chin et al., 2002; Kinne, 2019). The chemical aging scheme developed in this study (see Sect. 2.2.4) is the first to simulate aging through oxidant-driven mass-transfer between tracers of different optical properties, rather than the typical approach of parameterizing optical properties as a function of time in the atmosphere, allowing for the formation of



more complex, realistic OA mixtures. This is also the first study to account for browning, in addition to bleaching, in a chemical aging scheme. We estimated the radiative effect of BrC aerosols and performed sensitivity tests to determine the extent each defined BrC parameter changes this. Instead of a direct evaluation of BrC, which requires comparison to flight campaign and retrieval data, extensive work that will be presented in a future study, we evaluated general model performance with BrC aerosols. To do this, we compared simulated total aerosol optical depth and absorbing aerosol optical

depth to ground-based data from the Aerosol Robotic Network (AERONET) and satellite data from the Moderate Resolution Imaging Spectroradiometer (MODIS) instruments.

## 2 Model description and experiments

### 2.1 The GISS ModelE Earth System Model

This study used version 2.1 of the GISS ModelE Earth system model, ModelE2.1. The horizontal and vertical resolution of

the atmosphere in ModelE2.1 is 2º in latitude by 2.5º in longitude with 40 vertical layers from the surface to 0.1 hPa. ModelE includes multiple aerosol schemes (Bauer et al., 2020). We used the One-Moment Aerosol (OMA) module, because it includes more detailed OA chemistry. OMA is fully interactive within ModelE in terms of emissions, chemistry, transport, removal, and climate. The aerosol-radiation interactions (ARI) and aerosol-cloud interactions (ACI) are calculated within the radiation and cloud schemes, where the size-dependent scattering properties of clouds and aerosols are computed from Mie

scattering. To account for aerosol swelling with water vapor, relative humidity, aerosol hygroscopicity and the refractive index of water are used in the calculation of such properties. In ModelE aerosol scattering, asymmetry factor, and light extinction for six wavelength bands in the shortwave (SW) and 33 in the longwave (LW) are used to calculate ARI (Bauer et al., 2010). With regards to ACI, OMA only includes the first indirect effect (Bauer et al., 2020).

      OMA is a mass-based scheme in which aerosols are assumed to have prescribed and constant size distributions. Aerosol

components represented are sulfate, nitrate, ammonium, dust, sea salt, and carbonaceous aerosols. Carbonaceous aerosols include BC and OA, which are each separated into aerosols from industrial and BB sources. OMA also simulates the formation of biogenic SOA, discussed further in Sect. 2.2.2. Within the original ModelE radiation, industrial and BB OA, as well as SOA, are considered to have the same optical properties, with all organics treated as slightly absorbing in the UV-visible wavelength band. Sea salt, dimethyl sulfide (leading to methanesulfonic acid), isoprene (leading to SOA), and dust

emission fluxes are calculated interactively, while all remaining anthropogenic and BB fluxes are prescribed by the Community Emissions Data System (CEDS; Hoesly et al., 2018).

      This study made use of both climatological and nudged, transient simulations. Climatological simulations were used to assess model sensitivity to BrC representation (see Sect. 2.3.1) and utilized CEDS emissions for aerosols from BB, as used in CMIP6 (Hoesly et al., 2018). CEDS BB emissions are identical to the Global Fire Emissions Database version 4 (GFED;

van der Werf et al., 2017; McDuffie et al., 2020) for the year 1997-2014. Nudged, transient simulations were used to



compare model output to AERONET and MODIS retrieval data (see Sect 2.3.2) and utilized the Global Fire Assimilation System version 1.2 (GFAS1.2) for BB aerosol emissions (Kaiser et al., 2012). GFAS1.2 was used, rather than other fire emissions inventories, as it allows for implementation of plume injection height, rather than the ModelE default of all BB emissions injected uniformly in the boundary layer, and also has daily emissions, instead of the monthly in CEDS (Freitas et al., 2007; Sofiev et al., 2012). It should be noted that, on average, globally, GFED4 OA emissions have been shown to be lower than GFAS1.2. Regionally, GFAS1.2 showed higher emissions in the Temperate North American (TENA), Southern Hemisphere South America (SHSA), Boreal Asia (BOAS), Southeast Asia (SEAS) and Equatorial Asia (EQAS) BB regions (Pan et al., 2020). This resulted in higher OA emissions in our transient simulations compared to our climatological simulations (approximately 25.9 Tg yr$^{-1}$ vs 24.6 Tg yr$^{-1}$, on average). Transient simulations were nudged towards 3-hourly winds prescribed by Modern-Era Retrospective Analysis for Research and Applications, version 2 (MERRA-2; Gelaro et al., 2017).

## 2.2 Brown carbon scheme

To simulate BrC, we defined emissions, formation of secondary BrC, its optical properties and its chemical aging in the atmosphere. The following sections discuss our methodology for estimating parameters in each of these scheme components. Since BrC is a broad classification of aerosols, there is a large degree of variability in observed properties. As a result, the parameters we present here, though based on laboratory and field studies of BrC, are inherently uncertain.

### 2.2.1 Emissions

BrC was introduced as a new set of aerosol tracers into the OMA module of ModelE2.1. Primary BrC aerosols are emitted by attributing a proportion of BB emissions from OA to BrC. This is equivalent to assuming a certain proportion of BB OA are absorbing, rather than non-absorbing and completely scattering. This study used prescribed BB emissions from the CEDS fire emission inventory for sensitivity tests and GFAS1.2 for evaluation against AERONET and MODIS retrieval data. CEDS was employed in sensitivity tests for consistency with CMIP6, while GFAS was used for better accuracy of OA spatiotemporal variability (due to injection height and daily data, as previously mentioned) and therefore a better comparison with retrieval data. Though the current ModelE implementations of both emission inventories do not differentiate BB fuel types, the mass ratio of absorbing to non-absorbing, or BrC-to-OA, emissions will vary globally with different vegetation biomes. For instance, Jo et al. (2016) estimated that croplands have the highest BrC-to-OA mass ratio, between 0.4 and 0.946 depending on assumed aerosol AAE, woody savannahs and shrublands have the lowest, between 0.046 and 0.123, with forests falling somewhere in the middle (0.093-0.135 boreal, 0.088-0.211 temperate, and 0.128-0.312 tropical). To find a representative global value, we looked to emission ratios used by other BrC modeling studies in addition to estimating a ratio from CEDS emissions used in ModelE. Literature values of global average BrC-to-OA mass ratios vary between 0.15-0.92, with an approximate average of 0.35 (Feng et al., 2013; Wang et al., 2014; Jo et al., 2016; Zhang et al., 2020).



To determine our own value of BrC-to-OA mass ratio, BrC emissions ($E_{BrC}$) were parameterized as a function of the global mean BC-to-OA BB emissions ratio from the CEDS inventory, following equation 1 (Saleh et al., 2014) and equation 2 (Zhang et al., 2020):

$$k_{BrC,550\text{ nm}} = 0.016 \log_{10}\left(\frac{E_{BC}}{E_{OA}}\right) + 0.03925, \tag{1}$$

$$E_{BrC} = \frac{4\pi \cdot [k_{BrC,550\text{ nm}}] \cdot E_{OA}}{\rho \cdot 550\text{ nm} \cdot MAE_{BrC}(550\text{ nm})}, \tag{2}$$

where $E_{OA}$ and $E_{BC}$ are OA and BC emissions, $k_{BrC,\,550\text{ nm}}$ is the imaginary refractive index (RI) of BrC at 550 nm, $\rho$ is the aerosol density in g m$^{-3}$, and MAE is the mass absorption efficiency of BrC in m$^2$ g$^{-1}$–we use a value of 1 m$^2$ g$^{-1}$ (McMeeking, 2008; Jo et al., 2016; Zhang et al., 2020). Equation 1 expresses BrC imaginary RI as a function of $E_{BC}$ to $E_{OA}$ ratio because fires with higher modified combustion efficiency (MCE), and therefore greater BC emissions, have been shown to produce more absorbing OA (Saleh et al., 2014; Liu et al., 2020). As all organic absorption is attributed to BrC, equation 2 uses the imaginary RI and MAE to determine how much BrC emissions would be needed to account for this absorption. Using this, we calculated area-weighted global mean BrC emissions of 3.98e-13 kgC m$^{-2}$ s$^{-1}$. Given total OA emissions of 1.09e-12 kgC m$^{-2}$ s$^{-1}$, we got an average BrC-to-OA emitted mass ratio of 0.366. Since this is close to the average mass ratio used in other BrC modeling studies, we chose 0.35 as the proportion of BB OA emissions attributed to BrC, making up approximately 10% of total OA mass (0.11 Tg burden) in ModelE. This value served as a starting point from which we conducted model sensitivity tests, as described in Sect. 2.3.1. We also applied the parameterization described in equations 1 and 2 globally (see Fig. A1), looking at BrC-to-OA emissions ratio in each grid cell rather than the global average, and found that 0.15-0.55 (15-55% BB organic emissions are brown) captures the entire range of estimated ratios and should be explored in these sensitivity tests.

### 2.2.2 Secondary, biogenic brown carbon

The formation of SOA from biogenic VOCs (BVOCs) was already represented in ModelE prior to this work. Briefly, BVOCs such as isoprene and terpenes are oxidized by hydroxyl and nitrate radicals, and ozone. A two-product model is utilized to account for VOC and reactive nitrate (NOx) conditions in SOA formation. This results in two aerosol species from each biogenic precursor–isoprene and α-pinene (representing terpenes). The yield within a model grid cell of the two products changes with NOx-to-VOC ratio (Tsigaridis and Kanakidou, 2007).

To account for secondary BrC, the model radiation scheme was modified to consider the four biogenic SOA products separately from other OA. This ensured a distinct, non-zero imaginary RI could be assigned to each tracer, allowing them to be absorbing. The actual values of these RIs will be discussed in Sect. 2.2.3. With this configuration, brown SOA makes up approximately 50% of total OA mass (0.57 Tg burden). SOA formed from anthropogenic, aromatic precursors have also been shown to be absorbing in the atmosphere (Liu et al., 2016). Aromatic SOA are not yet represented in ModelE, since



they are small contributors to the global OA budget (Tsigaridis and Kanakidou, 2003), potentially creating a low bias in secondary BrC mass.

### 2.2.3 Brown carbon optical properties

Imaginary RI in the visible wavelength band was the key property used to distinguish BrC from other OA in ModelE radiative transfer calculations. Imaginary RI determines to what extent our BrC tracers are absorbing. Given the primary purpose of this study is to improve representation of aerosol absorption, the real RI of BrC, which drives aerosol scattering, was kept the same as that of OA ($n_{OA}$) in all ModelE radiation bands. Additionally, since BrC demonstrates limited absorption past 800 nm (Laskin et al., 2015), only the imaginary RI for BrC in the UV-visible (UV-VIS) radiation band

(300-770 nm) was modified; the optical properties of BrC in all other radiation bands are the same as OA. The use of wide radiation bands, rather than distinct wavelengths, in radiation calculations poses a limitation for BrC representation: in the current implementation, ModelE is not able to capture the spectral dependence of BrC absorption in UV-VIS wavelengths, so a direct comparison of AOD and AAOD values from the literature at UV/near-UV wavelengths, where BrC absorption is maximized, is not possible without assuming an AAE. Instead, ModelE simulates one, spectrally weighted average value in

the UV-VIS band, indicative of λ=550 nm. In terms of radiative flux and forcing, using a spectrally weighted average for BrC RI and calculating mean forcing across the wider UV-VIS band is approximately equivalent to defining BrC optical properties in narrower wavelength bins, within the band, and summing bin forcing contributions. This is because, while BrC absorption increases into the UV, solar irradiance is much lower at shorter wavelengths, so resolving BrC radiative flux in the UV has limited impact on that of the total band. Thus, the use of one spectrally weighted RI should not pose a limitation

for estimating the net BrC radiative effect.

Just as differing biomes produce different BrC-to-OA emission ratios, the imaginary RI of primary BrC varies with combustion conditions and fuel type (Fleming et al., 2020). In ModelE, however, only one imaginary RI can be defined for each wavelength band of a tracer for AOD and forcing calculations. We used two parameterizations to estimate a range of representative imaginary RI for BB BrC. The first parameterization consisted of two steps. First, it used the Kramers-Kronig

(KK) relations for a damped harmonic (Moosmüller et al., 2011) to compute spectra of real RI ($n$) and imaginary RI ($k$) for absorbing matter at each wavelength in the UV-VIS between 350 and 770 nm. These relations are given by

$$n = 1 + a \frac{v_0^2 - v^2}{\left(v_0^2 - v^2\right)^2 + (\gamma v)^2}, \tag{3}$$

$$k = a \frac{\gamma v}{\left(v_0^2 - v^2\right)^2 + (\gamma v)^2}, \tag{4}$$

where $a$ is a constant, $\gamma$ is a line width parameter, $v$ is the frequency of incident light ($c/\lambda$), and $v_0$ ($c/\lambda_0$) is the resonance frequency of the oscillator. Sumlin et al. (2018) show that these relations can reproduce measurements for the imaginary RI




of BrC peat smoke ($k_{BrC}$). Figure 1 provides such a fit for measurements of smoke emitted from smoldering combustion of Alaskan peatland (sample AK 4-8" 5% MC from their study). Also shown in Fig. 1 is the solar spectral irradiance of the UV-VIS band, for reference. Taking the solar spectrum weighted average of these $k_{BrC}$ values gave an imaginary RI value of
approximately 0.003 for the UV-VIS band.

Sumlin et al. (2018) also show that the KK relations alone underestimate the real RI, $n_{BrC}$, of BrC peat smoke. This problem can be solved by volume-mixing the KK results for the RI of smoldering peat smoke with the RI ($n_{HM}$) of a non-absorbing host matter (HM)–the second step in this parameterization. Choosing $n_{HM}$ = 1.50 with a volume-mixing ratio of $f_{HM}$ = 89% for our HM led to not only to a good fit of $n_{BrC}$ spectra, but it also retained the fit of $k_{BrC}$ spectra, in the UV-VIS
part of the spectrum. Furthermore, taking the solar spectrum weighted average of these $n_{BrC}$ spectra led to a UV-VIS averaged $n_{BrC} \approx 1.53$, equal to the ModelE default $n_{OA}$ This supports our assumption stated above that the real RI of BrC is kept the same as that of OA in all radiation bands of ModelE.

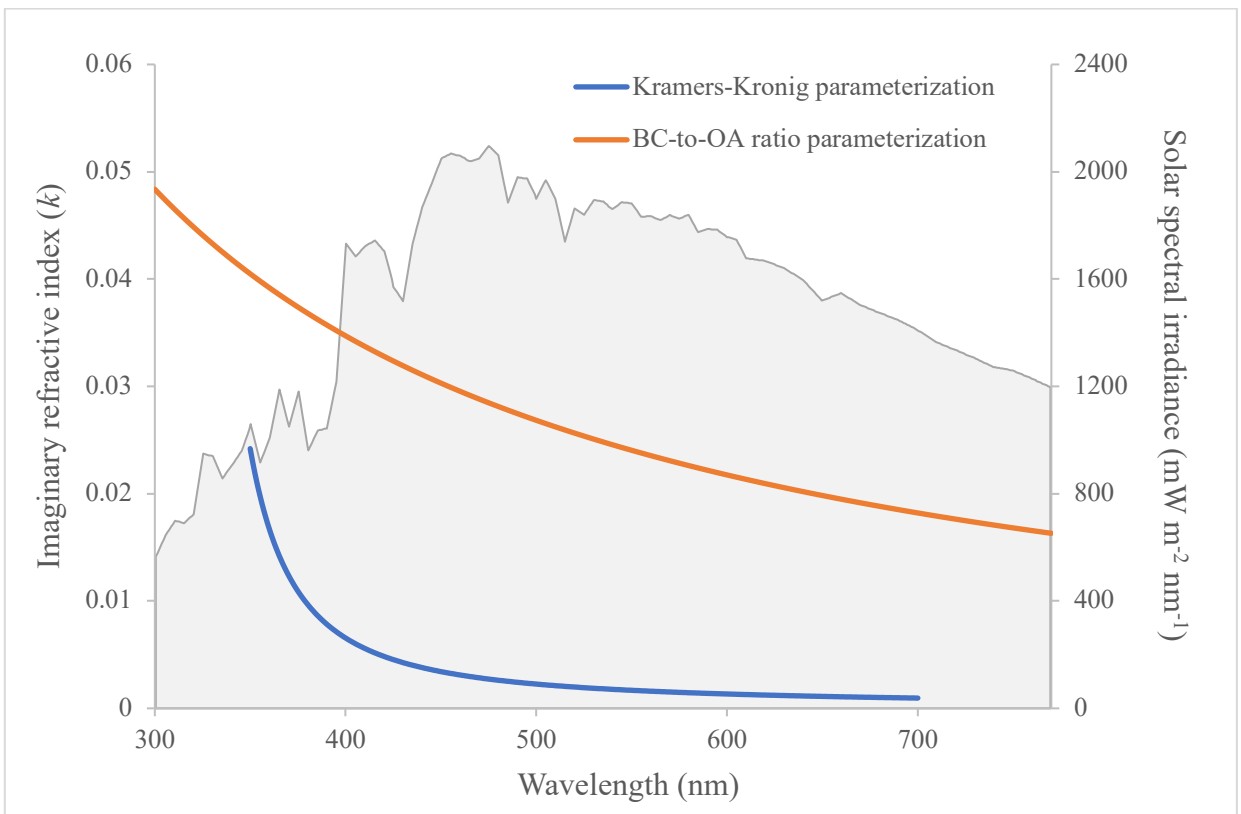

**Figure 1.** Imaginary refractive indices (RI) across 300 to 770 nm range for the Kramers-Kronig (KK) parameterization (blue) and BC-to-
OA ratio parameterization (orange) used to estimate primary BrC UV-VIS band imaginary RI. Note, data for the first parameterization is only provided from 350 to 700 nm. KK parameters used here are $a$=4.554e29 s$^{-2}$, $\gamma$=2.605e13 s$^{-1}$, $\lambda_0$=308 nm, and a BrC to non-absorbing host volume mixing ratio of 11%. These are applied to sample AK 4-8" 5% MC from Sumlin et al. (2018). The solar spectral irradiance of



the UV-VIS band is also shown here (grey) for reference. The derived imaginary RI of secondary BrC is not displayed here as the MAE data used was not continuous across this wavelength range and resulting RI values are much lower than that of primary BrC (<0.002).

The second parameterization is the same used to determine BrC emissions, where imaginary RI is a function of the BC-to-OA emissions ratio from the CEDS inventory (equation 1). We calculated a spectral absorption exponent (Lyapustin et al., 2021; Go et al., 2022), expressed below as w, according to equation 5 (Saleh et al., 2015):

$$w = \frac{0.21}{\frac{E_{BC}}{E_{OA}}+0.7} ,$$                                  (5)

Our estimate of w, which defines the spectral dependence of BrC absorption, was 1.15. This is close to the average value in
Saleh et al. (2014) of 1.6, and lower than other reported values of 3.9 (Kirchstetter et al., 2004) and 5.4-5.7 (Mok et al., 2016). Lower spectral dependence, using this parameterization, is correlated with higher BC-to-OA ratios and, therefore, higher imaginary RI (Saleh et al., 2014). This w value was then used to determine RI across all UV-VIS wavelengths (equation 6; Saleh et al., 2015):

$$k_{BrC}(\lambda) = k_{BrC,\,550\,nm} \bullet \left(\frac{550}{\lambda}\right)^{w} ,$$                                  (6)

The resulting imaginary RI can also be seen in Fig. 1. A solar spectrum weighted average of values from this calculation, following equation 5, gave an imaginary RI of approximately 0.03. Imaginary RI of both 0.003 and 0.03 are consistent with the range of values used by other BrC modeling studies, with $k_{BrC}$=0.003 representing weakly absorbing BrC at the bottom of the range, and $k_{BrC}$=0.03 representing strongly absorbing BrC at the top of the range, as expected with a low w value (Feng et al., 2013; Lin et al., 2014). A moderately absorbing case was also defined at the midpoint of this range, with $k_{BrC}$=0.0165.

For biogenic SOA, we used values of MAE for isoprene and α-pinene SOA measured under both high and low NOx conditions from Liu et al. (2016) to calculate the imaginary RI. The two SOA tracers for each BVOC in ModelE do not directly correlate with high and low NOx. Rather, each tracer has a specified mass yield given NOx conditions at the time and location of formation. We converted these tracer mass yields to molar yields, then solved a system of equations to determine the MAE of each tracer: for either isoprene or α-pinene, the MAE of a low or high NOx SOA (from Liu et al.,
2016) was set to equal the sum of the two tracers' low or high NOx molar yields multiplied by the respective tracer MAEs (solved for). The resulting MAE values were converted to imaginary RI according to equation 7 (Zhang et al., 2020):

$$k_{SOA}(\lambda) = \frac{MAE_{SOA}(\lambda) \bullet \rho \bullet \lambda}{4\pi} ,$$                                  (7)

Solar spectrum weighted averages across all UV-VIS wavelengths were taken, and the resulting imaginary RIs can be seen in Fig. 2, along with RIs for primary BrC and other aerosol tracers, for reference. Other BrC properties defined in ModelE
are listed in Table 1.




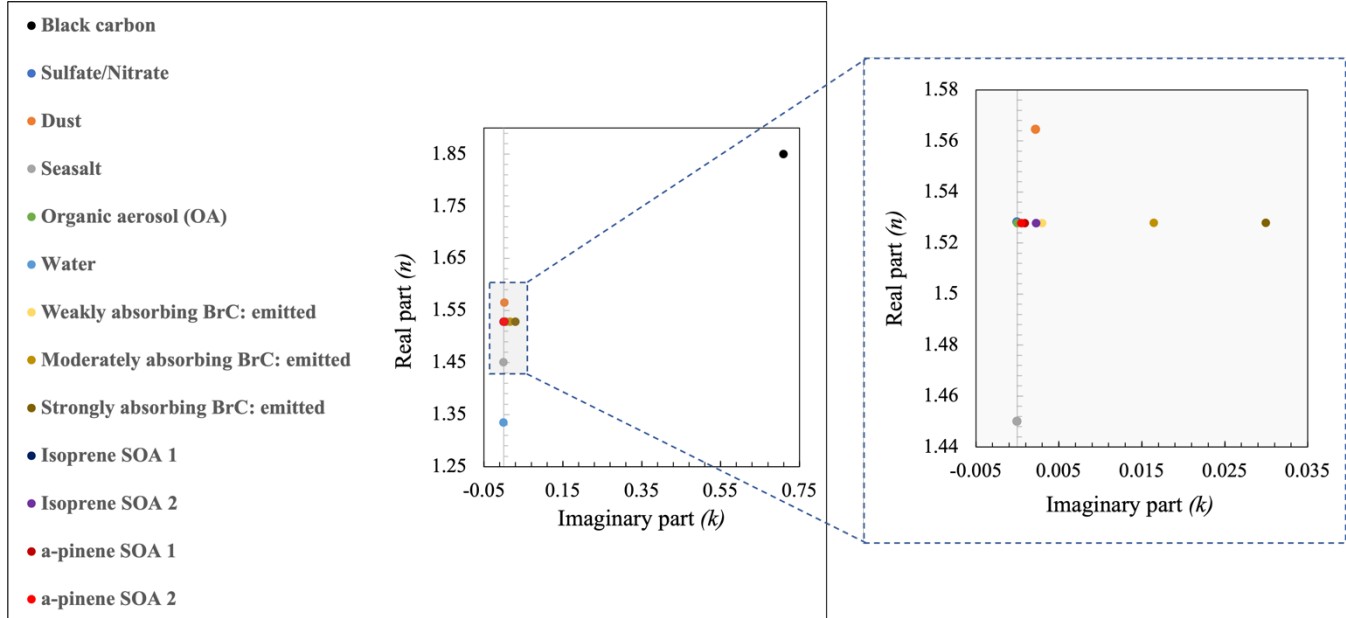

**Figure 2.** UV-VIS band averaged, complex refractive index of ModelE aerosol tracers. All BrC tracers have the same real refractive index as OA.

| Aerosol property | BrC value |
|---|---|
| Density | 1.5 g/cm$^3$ |
| Radius | 0.2 μm |
| Solubility (fraction of aerosol that dissolves) | 0.8 |
| Hygroscopicity (κ factor) | 0.15 |

**Table 1.** BrC physical properties used in ModelE. Each property is the same as other OA originally in ModelE (Koch, 2001). These values are consistent with estimates of BrC properties from previous laboratory studies and BrC reviews (Lin et al., 2014; Laskin et al., 2015; Froyd et al., 2019).

### 2.2.4 Chemical aging scheme

Since the objective of this work is to capture OA absorption, we focused only on changing BrC optical properties with aging. As such, all properties in Table 1 are kept constant. To simulate the atmospheric processing of BrC, we created an oxidant-initiated chemical aging scheme that does not require tracking the change in RI, and therefore absorption, over time. Instead, two aged BrC tracers were introduced in ModelE, in addition to the emitted one; one with higher absorption and one with lower absorption in relation to emitted BrC. The "browner" BrC is assumed to have 150% of emitted BrC absorption efficiency, while the less absorbing BrC is assumed to have 20%. These relative absorptions are based on laboratory data of



oxidized BrC proxies (Zhao et al., 2015), as well as threshold absorptions applied by other modeling studies (Wang et al.,
2018). Mie calculations were used to determine what imaginary RI, varied from that of emitted BrC, produce the relative
absorption efficiencies of each aged tracer. This was done for all three primary BrC cases described earlier–weakly,
moderately, and strongly absorbing.

BrC browning is represented as the transfer of emitted (primary) BrC mass to the more absorbing BrC tracer. This is
followed by bleaching, with mass transfer from the more absorbing to the least absorbing BrC. Mass transfer between tracers
occurs at a rate determined by a second order rate constant for each reaction of BrC with hydroxyl and nitrate radicals, and
ozone (see Table A2). We used first order rate constants defined by Hems et al.'s (2021) kinetic model of BrC processing
and their assumed concentrations of oxidants to calculate these second order rate constants. This chemical aging scheme is
illustrated in Fig. 3.

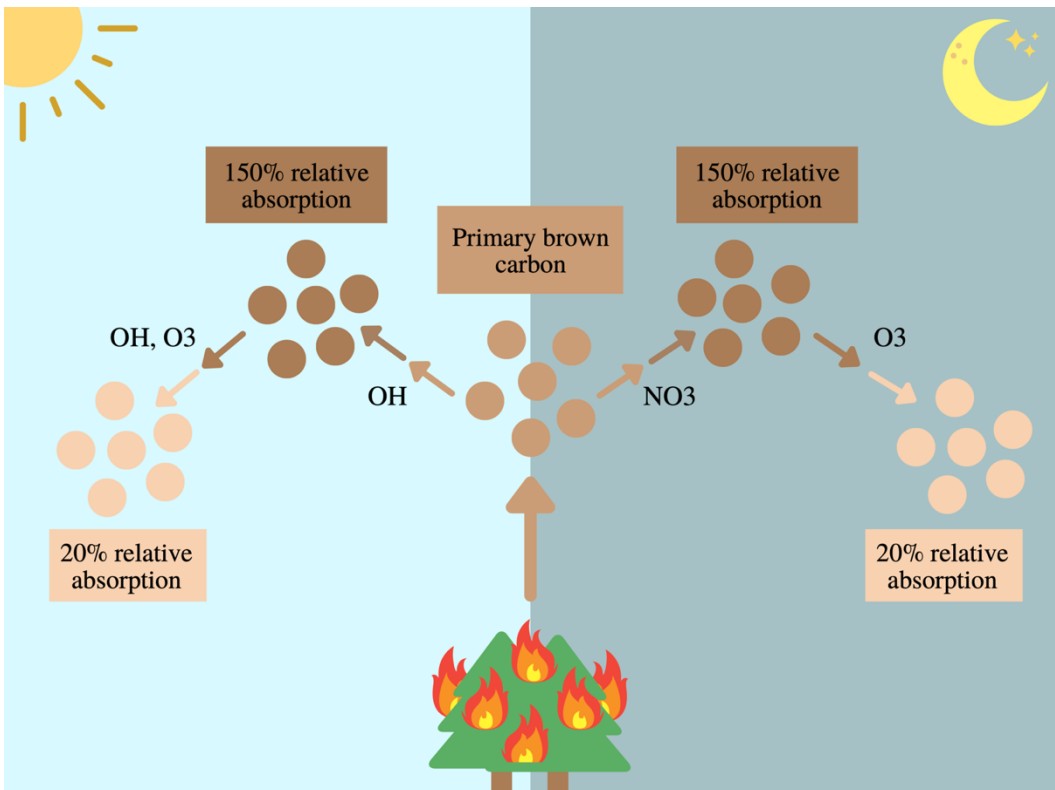

**Figure 3.** Chemical aging scheme of primary BrC in GISS ModelE. The large arrow represents emission of BrC mass from BB while
smaller arrows represent mass transfer between BrC types.

The typical chemical lifetime of ModelE simulated emitted and "browner" BrC in this scheme are 9.36 hours and 6.79
minutes, respectively, meaning the entire aging process occurs, on average, over about 10 hours. This is consistent in
magnitude with atmospheric lifetimes predicted by laboratory studies, with browning occurring over several hours and rapid
bleaching on the order of an hour or less (Zhao et al., 2015; Hems et al., 2021). Because hydroxyl radicals are only



prominent during the day, and nitrate radicals during the night, the separate consideration of each of these oxidants allows for an oxidant-based diurnal simulation of BrC aging. Considering only the reactions occurring at night (nitrate browning and ozone bleaching), based on first order rates from Hems et al. (2021), we'd expect emitted BrC to have a chemical lifetime of about 40 hours, and "browner" BrC a lifetime of about 18 minutes, compared to 14.6 hours and 17 minutes, respectively, for just daytime reactions. This greatly extends the length of the total aging process, which is consistent with literature that has suggested a slow build-up of more absorbing BrC overnight (Li et al., 2020a).

The total radiative effect of all three BrC tracers, considered together, represent the effect of BrC that has been emitted and then aged heterogeneously in the atmosphere. The design of this aging scheme is unique in global BrC modeling; no other studies have simulated browning, and all but one of those with bleaching parameterized BrC absorption to decay with time (Zhang et al., 2020), or time and hydroxyl concentration (Brown et al., 2018; Wang et al., 2018; Tuccella et al., 2020; Carter et al., 2021), rather than simulating different types of BrC tracers. Drugé et al. (2022) is the only other BrC modeling study to simulate aging through the transfer of aerosols between bins with different optical properties, though they used a set lifetime, while we used local oxidant concentrations to determine rate of transfer (aging).

Our aging scheme is missing two key processes. Firstly, it only represents heterogenous aging. While there is laboratory evidence that BrC also undergoes in-cloud processing (Hems et al., 2021), this has not yet been introduced in ModelE. The addition of in-cloud, also referred to as multi-phase, processing would likely accelerate the overall rate of BrC aging: according to Hems' kinetic model, including aqueous reactions would shorten browning lifetime to approximately 3 hours and have limited effect on bleaching lifetime. Secondly, biogenic brown SOA do not currently undergo aging, though studies suggest bleaching of absorbing SOA occurs in the atmosphere at similar rates as primary BrC (Zhao et al., 2015; Liu et al., 2016). Our current BrC aging scheme is incompatible with ModelE's biogenic SOA parameterization: BrC is aged by moving mass from one tracer to another, but this violates the two-product model that produces SOA (Tsigaridis and Kanakidou, 2007). A different approach to chemical aging, one in which the semi-volatile nature of the aerosol is considered, must be developed to account for secondary BrC bleaching. Without this, we may be overestimating SOA contribution to BrC absorption: laboratory studies suggest aging reduces SOA absorption by at least 50% (Liu et al., 2016). As such, we plan to include SOA aging in future work.

## 2.3 Model assessment

This BrC scheme was assessed in two ways. First, an investigation of the radiative effect of BrC and its uncertain parameters used in ModelE, defined in the previous section, was performed through sensitivity tests. Next, ModelE simulated total aerosol properties, with BrC representation included, were evaluated through comparison to AERONET and MODIS retrieval data. This latter evaluation serves to contextualize ModelE BrC-included simulations and broadly assess the model's ability to capture aerosol properties. The purpose of sensitivity tests of parameters and assessment of model performance for total aerosol properties was to understand the overall impact of BrC on ModelE ARI. With this understanding, ModelE with the BrC scheme can be used for more detailed, future studies of BB aerosols.





### 2.3.1 ModelE BrC sensitivity tests

We conducted sensitivity tests to quantify the radiative effect of BrC representation in ModelE as a function of a range of the uncertain parameters described in Sect. 2.2. The following BrC processes and parameters were investigated: BB emission fraction, inclusion of brown biogenic SOA, primary BrC optical properties and chemical aging of primary BrC. We varied these, changing just one property at a time, from what we consider the base case for representation: 35% of BB OA emissions are brown, biogenic SOA defined as brown, primary BrC with moderate absorption, and inclusion of both

browning and bleaching processes. We also ran two simulations where BrC was not explicitly represented, one in which all organics are considered non-absorbing (our control case), and one in which all organics are somewhat brown with a non-zero imaginary RI, as is the default case for organics in ModelE. The details of each simulation are included in Table 2.

| Simulation | BrC | Brown biogenic SOA | $k_{OA}$ | Primary BrC case | $k_{emitted}$ BrC | $k_{150\% abs}$ BrC | $k_{20\% abs}$ BrC | % OA BB emissions are brown | Aging processes |
|---|---|---|---|---|---|---|---|---|---|
| 1 (control) | No | | 0.0 | | | | | | |
| 2 (default) | Implicit | No | 0.00567 | | | | | | |
| 3 | | | | Moderate | 0.0165 | 0.0293 | 0.00266 | | |
| 4 | | | | Weak | 0.003 | 0.00463 | 5.75e-4 | 35% | Browning and bleaching |
| 5 (base) | | | | Moderate | 0.0165 | 0.0293 | 0.00266 | | |
| 6 | | | | Strong | 0.03 | 0.0653 | 0.00415 | | |
| 7 | Explicit | Yes | 0.0 | | | | | 15% | |
| 8 | | | | | | | | 55% | |
| 9 | | | | Moderate | 0.0165 | 0.0293 | 0.00266 | 35% | None |
| 10 | | | | | | | | | Bleaching |
| 11 | | | | | | | | | Browning |

**Table 2.** BrC representation parameters for each sensitivity test simulation, where $k$ is the imaginary RI of an aerosol in the ModelE UV-VIS radiation band (300-770 nm). Simulations 1-2 are the two cases in which BrC is not explicitly represented, with 1 being the control

case where no organics are brown and 2 being the current model default, where all organics are slightly brown. Simulation 5 is the base case for BrC representation, using parameters established in Sect. 2.2. Simulations 3 and 5 are identical except for their treatment of secondary BrC: either excluded (simulation 3) or included (simulation 5). Simulations 4 and 6 test the effect of changing primary BrC optical properties (compared to simulation 5). Simulations 7 and 8 (compared to simulation 5) test the effect of changing OA BB emission percentage considered brown. Simulations 9-11 (compared to simulation 5) test the effect of BrC chemical aging processes.



The purpose of this testing, in addition to estimating BrC's radiative effect, was to understand the relative importance of each of the BrC processes included in the model, and how sensitive model results are against a plausible range in each one of them.

All tests were run using climatological simulations representative of a decadal mean centered around the year 2000, using three years for spin-up and fifteen years for analysis. Results are reported as global averages over the 15-year analysis

period, with standard deviation serving as a metric of the internal variability of the model. We calculated the direct radiative effect of BrC as the difference between a simulation and the control, top-of-atmosphere (TOA) instantaneous radiative forcing (IRF). IRF is the difference between including a tracer in model radiation calculations and not, via double calls to model radiation, at every time step. This was done for each simulation $i$ (see Table 2) other than the control, following equation 8:

$$RE_{BrC,i} = IRF_{TOA,ARI,i} - IRF_{TOA,ARI,ctrl}, \tag{8}$$

where RE is the direct radiative effect. This definition of radiative effect should be distinguished from effective radiative forcing, also commonly reported in modeling studies, which is the present-day radiative effect of a tracer compared to its pre-industrial effect, allowing the atmosphere to adjust to perturbations from that tracer (Hansen et al., 2005).

We expect BrC aerosols to mainly impact ARI; ACI are likely only impacted marginally through the effect absorbing

aerosols have on atmospheric stability and clouds, referred to as semi-direct effects. The interaction of aerosols on clouds via cloud condensation nuclei (CCN) changes are simulated in the model, but since BrC maintains the same solubility and hygroscopicity as other organics, and since we did not add organic mass to ModelE, we do not expect BrC representation to change CCN and have an impact on ACI discernible from simulation noise. Additionally, only the UV-VIS radiation band (300-770 nm) direct effect was considered, as BrC refractive indices mainly differ from other organics in this band, and we

wouldn't expect a radiative effect beyond it.

### 2.3.2 Evaluation against global aerosol retrieval data

We evaluated the model's ability to capture total aerosol extinction and absorption when employing the new BrC scheme through comparison to retrievals of total aerosol optical depth (AOD) and absorbing aerosol optical depth (AAOD). This comparison was performed globally and regionally, in BB regions during peak fire months. We chose to focus on BB as it is

a key source of BrC (Chakrabarty et al., 2010; McMeeking et al., 2009) and initial model results showed OA, including BrC, are concentrated in BB regions (see Fig. 4).





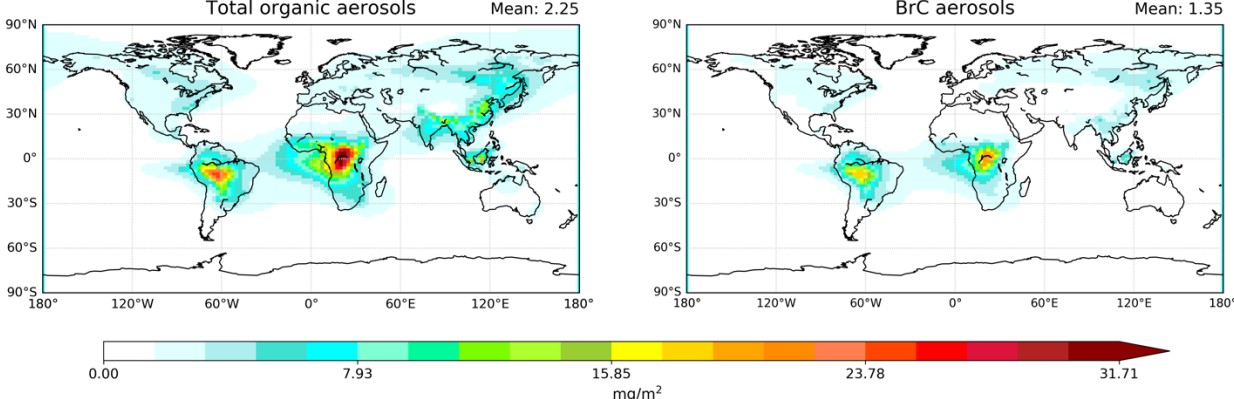

**Figure 4.** Annual average of total OA (left) and BrC aerosol (right) column burden under base case BrC representation, as described in Sect. 2.2. Both maps demonstrate organic and BrC aerosols are concentrated in BB regions of the Amazon, central/southern hemisphere Africa, and southeast and equatorial Asia. As BrC aerosols consist of biogenic SOA, in addition to BB emissions, high concentrations are also apparent in regions with high emissions of BVOCs. Similarly, industrial organic emissions contribute to total organic aerosol concentrations outside of BB regions. This map represents an annual average of a climatological simulation, which is a decadal mean centered around year 2000, so seasonal variations and emissions from BB regions that may have been prominent in certain years, like the western United States and Australia, are not visible.

To further determine the effect of BrC in ModelE, beyond Sect. 2.3.1, this comparison was performed against a control simulation in addition to a simulation of the base case of BrC representation, where BrC parameters used were taken from simulations 1 and 5 in Table 2, respectively. The goal was to determine if BrC representation changes model performance against retrieval data. Unlike sensitivity tests which make use of climatological simulations, this comparison was done with nudged, transient simulations using MERRA2 meteorology, to allow a better match to the actual observed period, and GFAS1.2 BB emissions, as stated in Sect. 2.1.

The Aerosol Robotic Network, or AERONET, consists of several hundred sun- and sky-scanning radiometers. Direct sun measurements and sky radiances are taken at typical wavelengths of 0.44, 0.675, 0.87, and 1.02 μm. AOD is a direct measurement product, while almucantar scans allow for size distribution and absorption retrieval products. We made use of Version 3, Level 2 (L2) inversion product data, which require, in addition to cloud screening, pairs of measurements with the same scattering angles to agree within 20%, at least 14 of these angular pairs to survive, and AOD at 0.44 μm (440 nm) to be greater than 0.4 for an AAOD retrieval to be considered. AAOD and AAE are linked to retrieved size-distributions and refractive indices through Mie theory or T-matrix theory and reported as column-integrated values (Sinyuk et al., 2020). The AOD measurements we used are coincident, meaning they were taken simultaneously with an almucantar scan.

We compared monthly averages of AERONET L2 AOD and AAOD over a ten-year period, 2007-2016, to ModelE simulated clear-sky, UV-VIS band optical depth. Since a solar-spectrum weighted average of wavelengths in the UV-VIS



band is approximately 550 nm, we used the AE and AAE provided in L2 data to calculate the AERONET optical depth values at 550 nm (Schuster et al., 2006). Monthly mean averages of AAOD were computed considering only months at a site with at least 10 days of daily averaged $AOD_{440\,nm} > 0.4$. Since retrieved AAOD values are highly uncertain with low AOD conditions, this aims to avoid considering months with too few, reliable AAOD measurements (Dubovik et al., 2000). In 400 addition to AERONET retrieval data, we compared ModelE simulated AOD to column AOD at 550 nm from the Moderate Resolution Imaging Spectroradiometer (MODIS) instrument on the Terra satellite, over the same ten-year period of 2007-2016. The benefit of using MODIS data, in addition to AERONET, is that it takes measurements over 36 spectral channels, allowing for better cloud screening and high accuracy over land and oceans (Levy and Hsu, 2015). We used Collection 6 Dark Target and Deep Blue combined product from the Terra satellite, with a resolution of 1° by 1°, for this analysis. The 405 Dark Target AOD product covers global oceans and dark surfaces of continents, such as vegetated areas, while the Deep Blue product includes retrievals over additional, brighter, land types (Levy et al., 2013).

When narrowing analysis to peak BB regions and months, we looked at Southern Hemisphere South America (SHSA), Southern Hemisphere Africa (SHAF), Temperate North America (TENA), Boreal North America (BONA), Southeast Asia (SEAS), Boreal Asia (BOAS), Equatorial Asia (EQAS) and Australia (AUST) (Fig. 5).

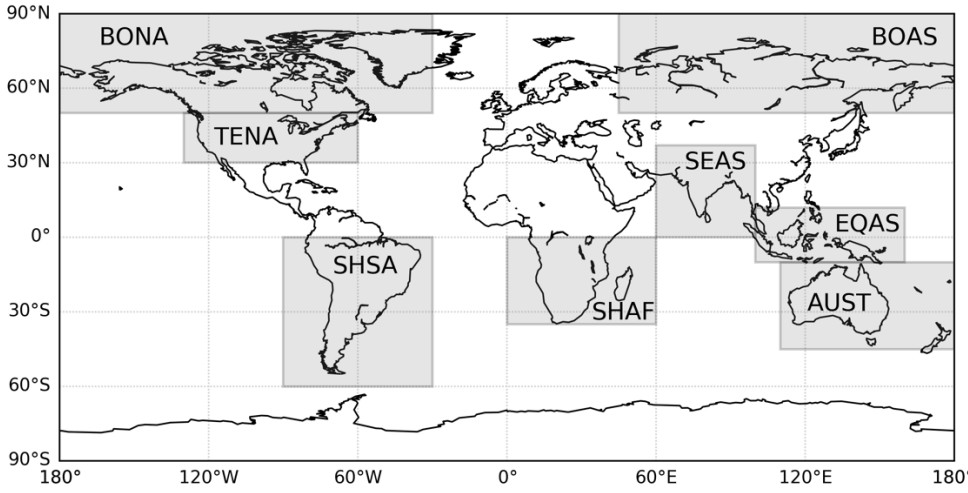

**Figure 5.** Map showing the eight BB regions used in this study, following regionalization defined in Pan et al. (2020).

The peak fire months for these regions are, broadly, boreal spring for BOAS and SEAS, boreal summer for TENA and BONA, and austral spring for SHSA, SHAF, EQAS, and AUST (Pan et al., 2020). The exact months considered to be peak fire periods for each region of analysis can be found in Table A2. We did not consider the Northern Hemisphere Africa 415 region in this focused analysis as any AOD comparisons would be strongly affected by dust, making the interpretation of BB-only results difficult.





## 3 Results and discussion

### 3.1 Changes in aerosol absorption with BrC parameterization

We used global averages of aerosol single-scattering albedo (SSA) from the model sensitivity tests to assess the effect of
BrC representation on aerosol absorption in ModelE; lower SSA indicates more absorbing aerosols. There is almost no change–a decrease of 0.001–in the global average total aerosol SSA with the introduction of BrC aerosols, which is expected since global absorption is dominated by BC and dust. Additionally, there is limited observable change in the spatial distribution of total SSA, and therein the spatial distribution of total AOD and total AAOD, across all sensitivity tests (see Fig. 6).

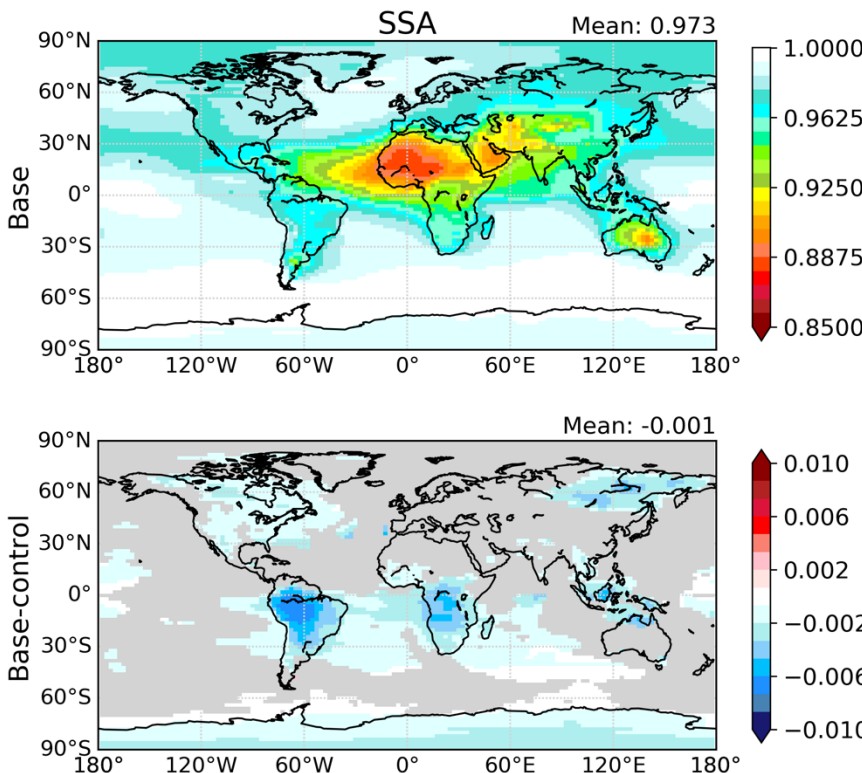


**Figure 6.** (Top) Total aerosol SSA in ModelE climatology simulation for base case of BrC representation (Sim. 5 in Table 2). Global distribution of SSA in the control case, with no BrC simulated and all organics treated as non-absorbing (Sim. 1 in Table 2), is not pictured as it appears identical to that of the base case. (Bottom) Difference in total SSA between base and control case simulations. Only data at 95% confidence level or higher, with differences attributable to changes in OA treatment rather than random noise, are shown–remaining
data is greyed out. Though there is no apparent change in spatial distribution of total SSA, there are small changes in SSA magnitude in regions where BrC and OA aerosols are highly concentrated (see Fig. 4).




While BrC has a limited effect on total aerosol absorptivity, it does influence total OA absorption. In general, total OA SSA decreases with more absorbing organics, due to either greater amount or more absorbing BrC simulated. This can be seen in Fig. 7: compared to the control case where no BrC is represented and all OA is considered non-absorbing, SSA

decreases when either a) secondary BrC is included; b) primary BrC changes from weakly to strongly absorbing; c) BrC-to-OA BB emissions ratio increases; or d) primary BrC aging is excluded.

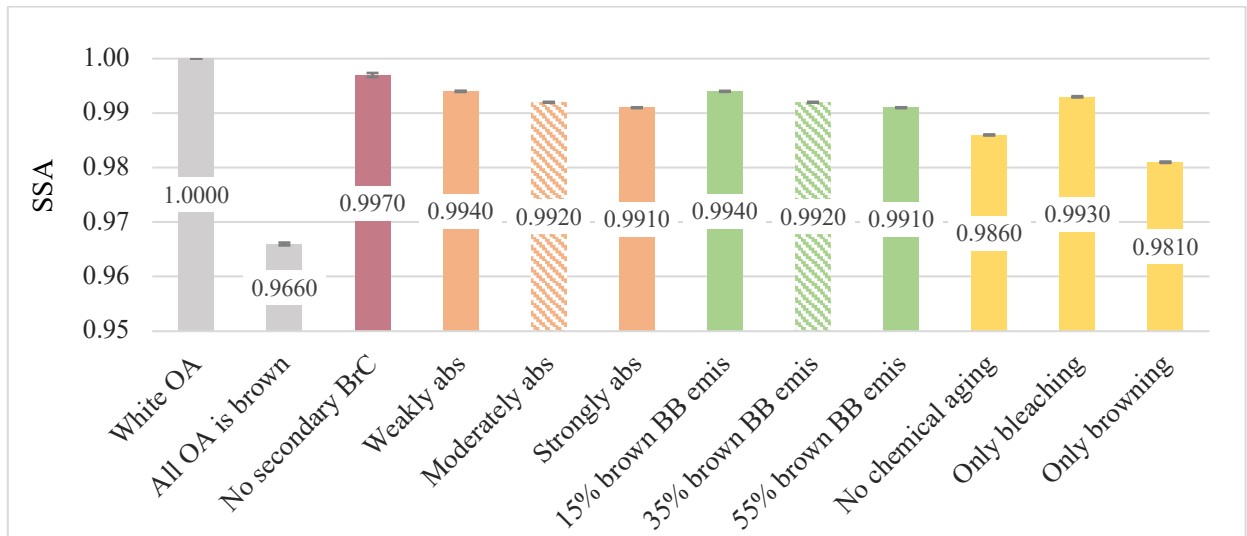

**Figure 7.** Global, annual average of total OA SSA across sensitivity test simulations. The dashed bars show the base case simulation (this is shown twice for ease of comparison to other simulations). The effect of each varied property can be seen by comparing the simulated

SSA to those of the control case and the base case. Error bars show the standard deviation of OA SSA and can be interpreted as the variability in each 15-year-long simulation. (Grey) Two simulations–control and default–where BrC is not explicitly represented. (Red) Properties and processes consistent with the base case except secondary BrC is not included. (Orange) Consistent with base case except for primary BrC RI which varies between weakly, moderately, and strongly absorbing cases. (Green) BrC-to-OA emissions ratio is varied, increasing from 15% to 55%. (Yellow) Primary BrC chemical scheme is varied, with no aging, only bleaching, and only browning

simulated.

For the latter case, considering only bleaching leads to the highest SSA of all aging sensitivities, because there is no "browner" BrC simulated. Additionally, since bleaching is a much faster process than browning, excluding browning allows primary BrC to move to the bleached state quicker: the chemical lifetime of primary BrC is reduced from 9.36 hours to just 10.3 minutes. On the diurnal timescale, this is like BrC just bleaching in the daytime, rather than building-up and browning

over several hours during the night.

The model's default case shows that assuming all OA is brown, where we do not separately represent BrC and apply one non-zero imaginary RI to all OA, results in the largest decrease in OA SSA. This default case, as well as the simulations with only browning as the chemical aging process, is not atmospherically realistic; only a fraction, not all, of OA have been observed to absorb light, and that absorbing portion has been observed, both in lab and field studies, to bleach (Cubison et





al., 2011; Laskin et al., 2015; Junghenn Noyes et al., 2020, 2022; Hems et al., 2021). We include these cases in our analysis, in addition to all other sensitivity test simulations, to bound BrC uncertainty.

## 3.2 BrC radiative effect

In the base case simulation, ModelE total OA radiative forcing (RF) is $-0.42 \pm 0.01$ W m$^{-2}$. This can be compared against the organic RF of the control and default cases of $-0.46 \pm 0.01$ W m$^{-2}$ and $-0.32 \pm 0.01$ W m$^{-2}$, respectively. The introduction of

absorption does not shift organic aerosols from negative to positive RF, since BrC is a weak absorber when integrated across a wide wavelength range. Instead, representation of BrC or the attribution of absorption to all organics result in a reduction of the total organic cooling effect. The direct radiative effect of BrC in the base case simulation, calculated according to equation 8, is $0.04 \pm 0.01$ W m$^{-2}$. For reference, Table 3 shows the TOA, instantaneous direct RF of other ModelE simulated aerosols.

| Species | Shortwave RF (W m$^{-2}$) | Longwave RF (W m$^{-2}$) | Net RF (W m$^{-2}$) |
|---|---|---|---|
| OA (control) | -0.49 | 0.03 | -0.46 |
| OA (default) | -0.35 | 0.03 | -0.32 |
| Sulfate | -1.25 | 0.06 | -1.19 |
| Sea salt | -1.25 | 0.01 | -1.24 |
| Nitrate | -0.09 | 0.0 | -0.09 |
| BC | 0.25 | 0.0 | 0.25 |
| Dust | -0.29 | 0.31 | 0.02 |

**Table 3.** Global annual average instantaneous TOA direct RF of ModelE aerosol species. Net RF was calculated as the sum of shortwave and longwave forcings.

BC aerosols have a large, positive RF (all from the shortwave), while dust aerosols have large shortwave and longwave forcing that are opposite in sign, resulting in a low net RF. These absorbing aerosols can be compared to BrC, which contributes a relatively small radiative effect. Figure 8 shows the global, spatial distribution of this BrC effect, which is

consistent with maps of total organic and BrC aerosols (see Fig. 4).



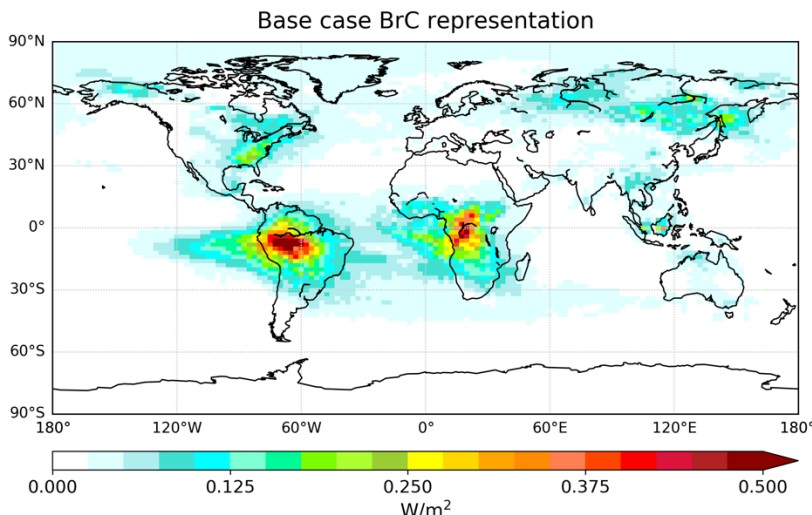

**Figure 8.** Annual average of BrC radiative effect in W m$^{-2}$, calculated according to equation 8.

### 3.2.1 Comparison of BrC radiative effect with previous studies

Our estimate of BrC radiative effect can be compared to similar BrC modeling studies. A wide range of radiative effects are reported across previous studies because of the variability in the treatment of BrC/OA absorption. Further, different studies use different metrics to quantify BrC's impact on Earth's radiative budget, with some reporting radiative effect as we have calculated, and others reporting instantaneous or effective RF. It's important to keep these variable treatments and climate metrics in mind as they pose a limitation to direct comparison between modeling studies.

As stated previously, there are three studies that have implemented BrC in an Earth system model. Two of these studies, using CESM (Brown et al., 2018) and CNRM (Drugé et al., 2022), calculated BrC effective radiative forcing of ARI (ERF$_{ARI}$). Brown et al. (2018) calculated an ERF$_{ARI}$ of 0.13 W m$^{-2}$ without BrC bleaching and 0.06 W m$^{-2}$ with bleaching, while Drugé et al. (2022) reported 0.029 W m$^{-2}$ with bleaching. The lower ERF reported by Drugé et al. (2022) may be a result of the different treatment of BrC-to-OA fraction: Brown et al. (2018) treated all BrC and OA as the same, while Drugé et al. (2022) defined BrC as BB OA and fossil fuel OA as non-absorbing. Both studies used the same parameterization for imaginary RI (see equation 1), with a global average $k_{550\,nm}$ around 0.02. Though we can't directly compare these studies to ours, as they calculated ERF rather than an IRF, our estimated BrC effect of 0.04 W m$^{-2}$ is similar in magnitude, with small differences likely attributable to differing BrC treatments. Brown et al. (2018) had a slightly larger global average BrC burden of 1.56 mg m$^{-2}$, compared to ours of 1.35 mg m$^{-2}$, a larger imaginary RI compared to ours of 0.0165, and a subsequently larger BrC effect. While Drugé et al. (2022) similarly used an imaginary RI larger than ours, they did not consider SOA to be brown and reported lower BrC emissions of approximately 1.73 Tg yr$^{-1}$ compared to ours of 8.6 Tg yr$^{-1}$ (or 6.14 TgC yr$^{-1}$, given the ModelE OA to organic carbon ratio of 1.4), resulting in a lower estimate of BrC effect.



Zhang et al. (2020) is the third study to use an ESM, CESM in particular, and presented the most similar approach to ours: primary BrC was considered a fraction of OA, SOA was considered brown, and a photobleaching parameterization was used. They also used the same approach to calculate BrC direct radiative effect, allowing for a direct comparison to ours.
Their treatment of BrC differed in that they used a BrC-to-OA emission factor of 23%, rather than 35%, considered aromatic SOA brown, rather than biogenic SOA, and used a higher imaginary refractive index of $k_{BrC, 550\ nm}$= 0.045, rather than our moderately absorbing case of $k_{BrC, 550\ nm}$= 0.0165. With this treatment, Zhang et al. (2020) calculated a BrC radiative effect of 0.1 W m$^{-2}$, which is larger than our base case estimate of 0.04 W m$^{-2}$. This is likely due to the higher imaginary refractive index applied to both primary and secondary BrC: they reported similar BrC emissions of 6.7 TgC yr$^{-1}$, and while our
production of brown SOA is much larger than theirs (16.1 vs. 4.1 TgC yr$^{-1}$), all ModelE SOA have imaginary RI less than 0.002, much lower than their singular RI used.

Other studies have estimated BrC instantaneous radiative effect using chemical transport models (CTMs)–either GEOS-Chem or IMPACT. Though these studies all differ in their treatment of BrC, they can be grouped according to whether they consider all OA brown or treat BrC as a fraction of OA. Studies that do not differentiate between BrC and other BB OA
report TOA BrC radiative effect between 0.048-0.57 W m$^{-2}$ (Lin et al., 2014; Saleh et al., 2015; Wang et al., 2018), while studies that treat BrC as a fraction of OA report between 0.04-0.29 W m$^{-2}$ (Park et al., 2010; Feng et al., 2013; Wang et al., 2014; Jo et al., 2016; Tuccella et al., 2020; Carter et al., 2021). Our calculated radiative effect is at the lower end of this reported values range.

### 3.2.2 ModelE sensitivity to BrC parameterization

In all sensitivity cases, BrC representation produces a reduction in organic cooling, or an effective warming. The actual magnitude of this effect, however, varies across simulations. Figure 9 shows the direct radiative effect and variability, expressed as standard deviation, of each BrC simulation.

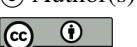





**Figure 9.** Global, annual average radiative effect of each BrC simulation, calculated according to equation 8. Error bars show the standard deviations, which can be interpreted as the variability of each simulation across repeated years of simulation. Different colored bars indicate a different BrC property varied, and dashed bars indicate the base case of BrC representation (this is shown twice for ease of comparison to other simulations), consistent with Fig. 7.

Comparing the no secondary BrC case to the base case, we can see that attributing absorption to organic SOA has a clear warming effect, since the magnitude nearly triples. Excluding chemical aging processes or only including browning also have a strong warming effect, compared to the base case. Changing the optical properties of primary BrC, either to the weakly absorbing or strongly absorbing case, varying the BB BrC-to-OA emissions ratio, and including only bleaching rather than both browning and bleaching do not produce distinctly different radiative effects from the base case. Finally, the default case where all OA are slightly brown shows substantial warming relative to the base case where only some organics are considered brown. Figure 10, showing BrC AOD and AAOD across the same simulations, indicates that while AAOD is much smaller in magnitude, it is clearly the larger driver in changing simulation radiative effect. There is also larger internal





variability in BrC radiative effect, compared to BrC optical depth, which we attribute to variability in simulated meteorology.

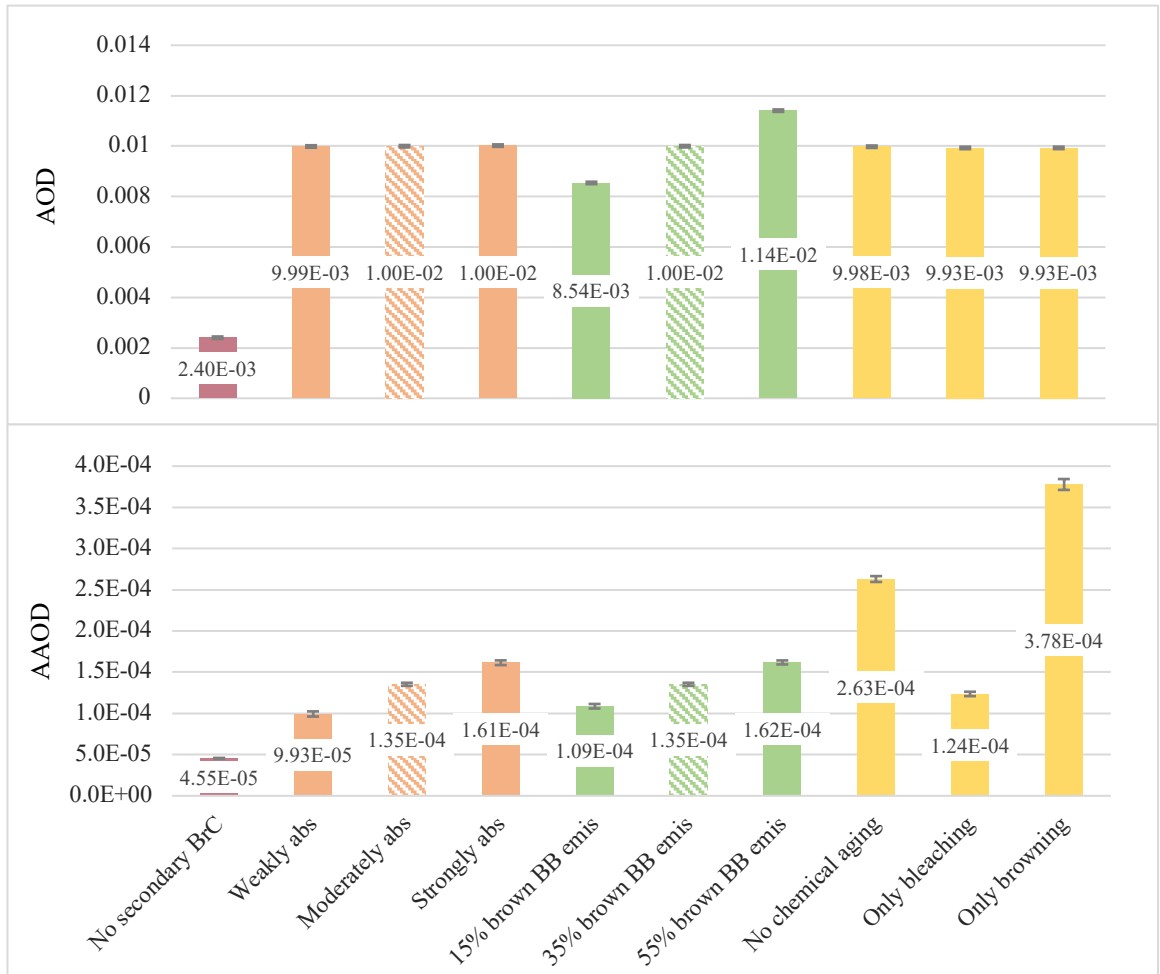

**Figure 10.** (Top) Total BrC AOD across each sensitivity test. (Bottom) Total BrC AAOD. Error bars show the standard deviations
(variability of each simulation across repeated years of simulation), different colored bars indicate a different BrC property varied, and dashed bars indicate the base case of BrC representation (shown twice for ease of comparison to other simulations), consistent with Figs. 7 and 9. ModelE default case is not displayed here as it does not explicitly simulate BrC, therefore no BrC optical depth could be calculated.

The sensitivity analysis presented in Fig. 9 shows that separating BrC from other organics through explicit representation, including secondary BrC, and simulating a chemical bleaching process all have a distinguishable effect on
ModelE BrC warming. Thus, each of these properties should be accounted for in BrC representation, and they should be the primary target for future BrC lab and field research to better constrain them. Since the base case BrC chemistry, browning followed by bleaching, is indistinguishable from the only bleaching case, simulating browning appears unnecessary on the scale of global annual averages, if the only interest is BrC radiative effect. Additionally, variation in primary BrC refractive





index and BrC-to-OA emissions ratio do not show distinguishable effects, suggesting it is not critical to define precise values
for these properties.

Since BrC forcing has a strong spatial inhomogeneity (see Fig. 8), the analysis demonstrated in Fig. 9 was repeated
within the BB regions and seasons discussed in Sect. 2.3.2. For examples of this analysis in the SHSA and AUST regions,
see Figs. A2-3. Within BB regions, the BrC radiative effect across all test cases is larger, appearing to scale linearly from the
global, annual effect. This makes sense, given BrC aerosols are more highly concentrated near BB sources and low
elsewhere (see Fig. 4). Further, since there is no regional difference in defined BrC physical or optical properties, an effect
proportional to the global average would be expected, with minimal differences resulting from regional SOA and oxidant
concentrations (affecting concentration of secondary BrC and rate of primary BrC aging). It should be noted that narrowing
the spatial and temporal scales of analysis also results in larger internal variability. As such, sensitivity tests are not
distinguishable from one another, and no additional conclusions can be drawn from this regional analysis.

**3.3 Evaluation of ModelE against retrieval data**

Figure 11 shows the comparison of ModelE simulated aerosol optical properties, with and without the BrC scheme, against
AERONET retrieval data. As described in Sect. 2.3.2, these are nudged, transient simulations with interannual variability,
rather than climatological, and used GFAS1.2 BB emissions, rather than CEDS. This scatterplot comparison, as well as the
linear regression analysis accompanying each plot, was done in the $\log_{10}$ space, rather than the linear space, as AOD is
known to be approximately log-normally distributed and any statistical analysis should reflect that (O'Neill et al., 2000;
Sayer and Knobelspiesse, 2019).





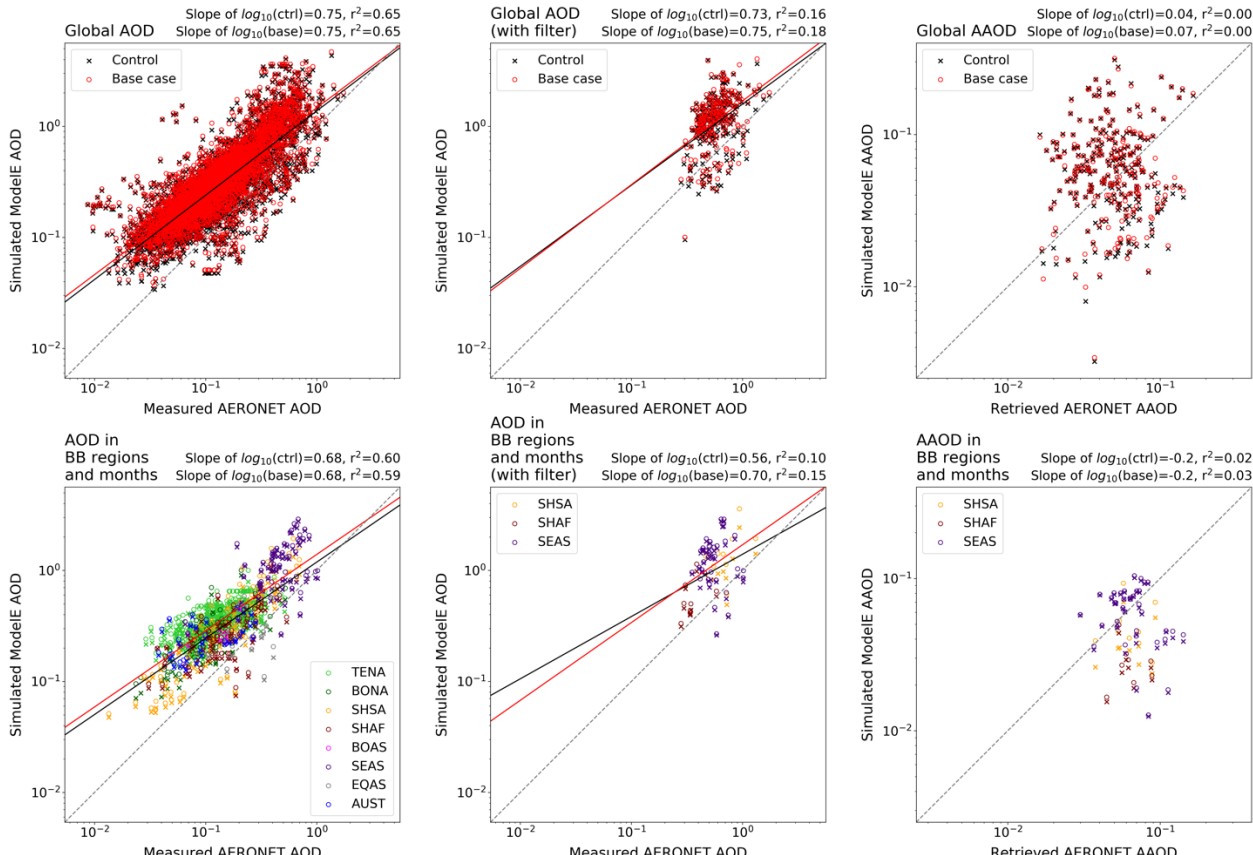

**Figure 11.** Simulated ModelE optical depth at 550 nm plotted in the $\log_{10}$ space against retrieved AERONET optical depth. Each point corresponds to the optical depth of one month, averaged across the 2007-2016 period, at an AERONET site and in the corresponding grid cell in ModelE. ModelE control case optical depth values are shown as 'x's, while base case values are shown as 'o's. The slope and $r^2$ of the linear regression are displayed on the top-right corner of each plot, and regression lines are included for all AOD plots. All AOD included are coincident. (Top left) AOD values at all available AERONET sites. (Bottom left) AOD in BB regions during months considered peak for BB, with each color representing a different region. (Top center) AOD at all available AERONET sites after $AOD_{440\ nm} < 0.4$ were removed. This is included to show corresponding AOD at the sites available for AAOD analysis (Bottom center) AOD in BB regions and months with $AOD_{440\ nm} < 0.4$ removed. Note TENA, BONA, BOAS, EQAS and AUST regions have been eliminated because of the $AOD_{440\ nm}$ threshold. (Top right) AAOD values at all available AERONET sites after months with less than 10 days of $AOD_{440\ nm} > 0.4$ were removed. (Bottom right) AAOD, with the same filter applied as the top right plot, in BB regions and months. As in the bottom center plot, five BB regions have been eliminated

Across all six plots, there appears to be limited difference between the ModelE control case and base case simulated optical depth, shown as x's and o's respectively. Linear regression analysis for both AOD and AAOD (their $\log_{10}$ values), on both the global and BB region scales, show minimal to no change in regression slope and $r^2$ value when BrC is explicitly



simulated. This suggests similar model skill against AERONET, with or without BrC representation. Further, this supports results discussed in Sect. 3.1: total AOD and AAOD were found to have no apparent change across all sensitivity tests, including the control case. The lack of a change in total AOD with the addition of BrC is not surprising, as no new aerosol

mass was introduced in the model. Additionally, OA and BrC have the same real RI, therefore scattering remains largely the same. Limited change in total AAOD is also expected, as demonstrated in Fig. 12, as total AAOD is usually dominated by either BC or dust aerosols.

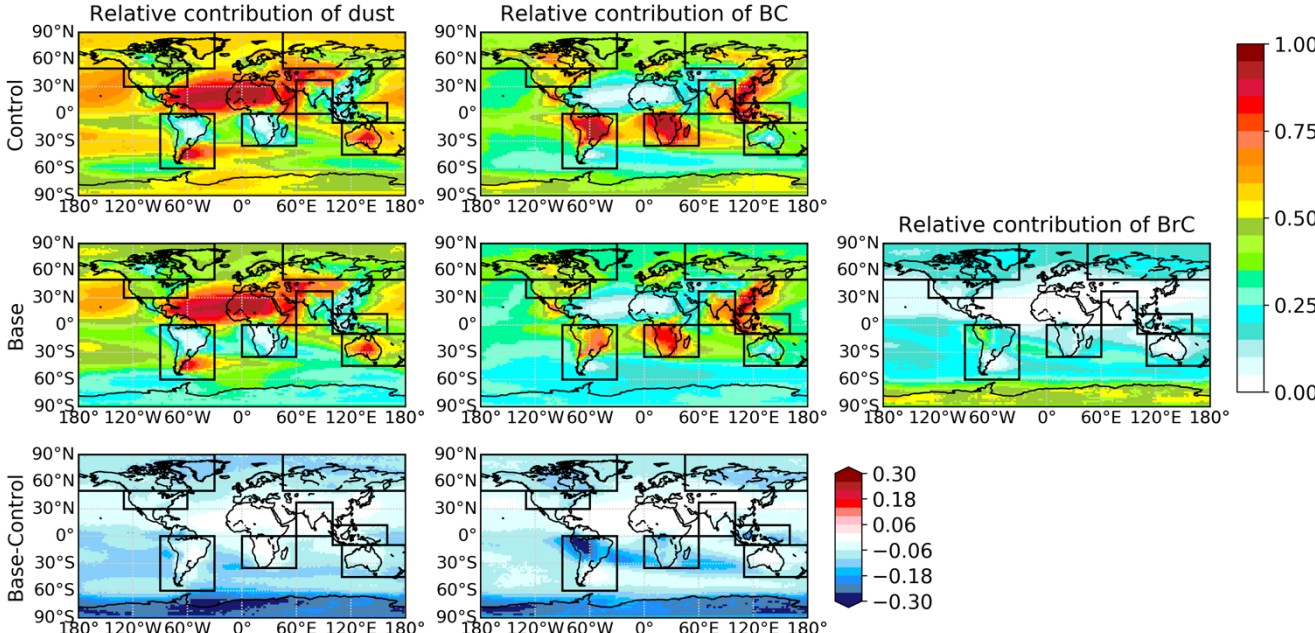

**Figure 12.** Average over the 2007-2016 period of ModelE light-absorbing dust (left), BC (middle), and BrC (right) contribution to total
AAOD in control (top) and base (middle) simulations, with each BB region of interest, identified in Sect. 2.3.2, outlined in black. (Bottom) Difference in dust and BC contributions between base and control cases. Relative contribution of BrC in BB regions appears to come from a greater reduction in that of BC, rather than dust.

Of the BB regions that we've focused our analysis on, AAOD is dominated by BC in most (SHSA, SHAF, TENA, BONA, BOAS, and EQAS), while dust dominates in AUST, and both BC and dust dominate in SEAS. BrC seems to account
for the majority of AAOD only over the Antarctic, where brown SOA in the remote free troposphere may be contributing more than dust or BC to the near-zero aerosol absorption occurring (Hu et al., 2013; Sand et al., 2017). It bears reminding that this comparison used retrieved AAOD at 550 nm, as that is the indicative wavelength of the ModelE UV-VIS wavelength band (see start of Sect. 2.2.3). If we were able to resolve BrC absorption and total AAOD within ModelE at shorter wavelengths, for instance 365 nm, we would likely see BrC have a much larger relative contribution to total AAOD.
This would particularly be the case over the BB regions currently dominated by spectrally flat BC absorption. Therefore, it is





possible that model performance between control and base case simulations would differ if we looked at UV/near-UV wavelengths, something we're currently unable to do within the ModelE radiation scheme.

Returning to Fig. 11, the left column demonstrates that, on both a global scale and within BB regions and months, ModelE tends to overestimate AOD relative to AERONET measurement, with greater overestimates at lower AOD values.
The center column of Fig. 11 shows the same AOD data as the left column with the data coverage filter for AAOD applied: all AOD values at 440 nm below 0.4 were removed. With this filter, data from TENA, BONA, BOAS, EQAS and AUST are lost, along with the strong linear relationship between retrieved and simulated data. Despite this, we can still see the ModelE overestimation of AOD, as most data points fall above the one-to-one line. Finally, looking at the right column of Fig. 11, the AERONET and ModelE AAOD comparison shows a large spread of AAOD values with no apparent linear relationship.
Though the model appears to underestimate AAOD in the SHAF region, as well as some sites in SHSA and SEAS, the limited sites with data make it difficult to draw any meaningful conclusions. Further, this data scatter is mostly caused by dust and BC, rather than BrC.

Figure 13 shows the global distributions of the ModelE bias in AOD and AAOD, relative to AERONET, for the months of March and August.




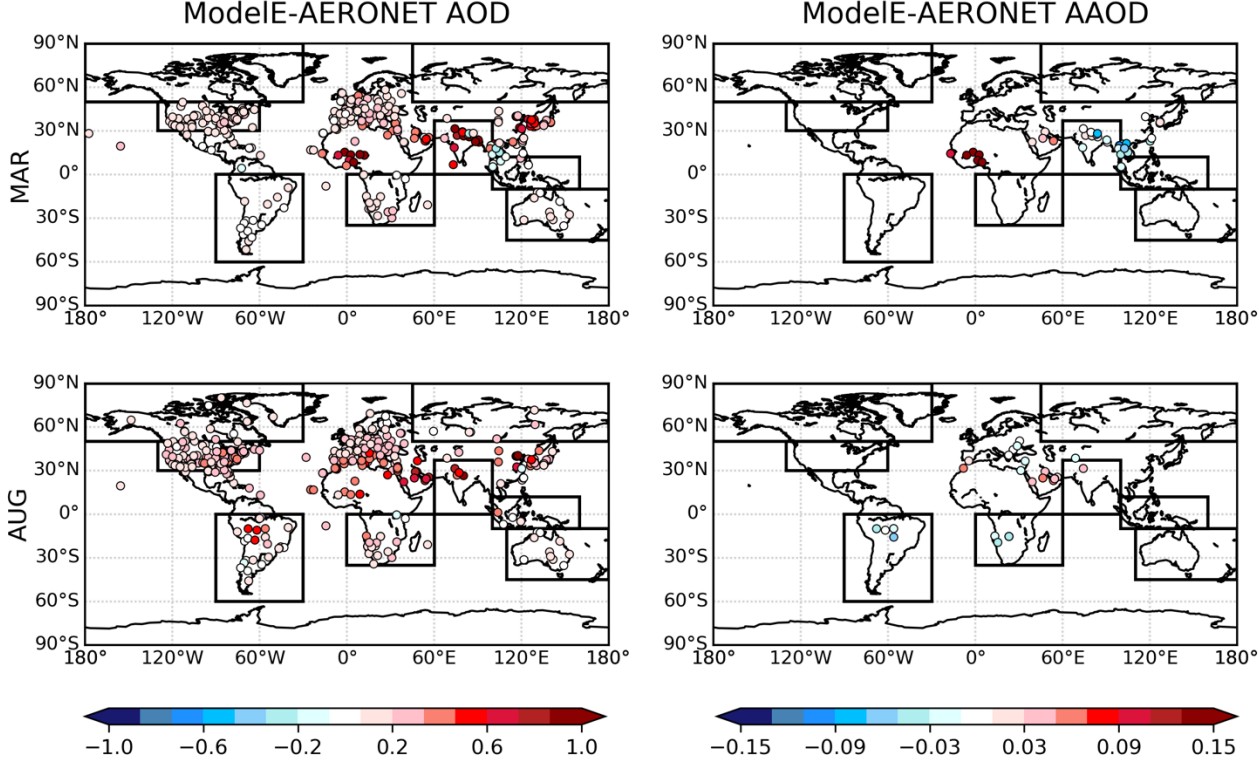


**Figure 13.** Map of model optical depth bias, where each point corresponds to the difference in ModelE base case and AERONET optical depth at an AERONET site, averaged oved the 2007-2016 period, for the months of March (top row) and August (bottom row). March and August are displayed as, together, they overlap with the BB season of almost all regions of interest.

This confirms the findings of Fig. 11: AOD bias maps show a global pattern of overestimation. While there is a limited
number of sites in the AAOD bias maps, there does appear to be negative bias at some sites in the SHAF, SHSA, and SEAS
BB regions. Again, due to the sparse data, no definitive conclusions can be drawn from this.

Evaluation of ModelE optical depth against MODIS retrieval data supports the conclusions from that of AERONET.
Firstly, there is no apparent difference between ModelE control case and base case simulated AOD when compared to
MODIS, confirming that model skill is unchanged with BrC representation (see Fig. A4). ModelE also tends to overestimate
AOD relative to MODIS, consistent with AERONET results. A comparison to MODIS data gives a clearer understanding of
this model bias, as we get a global picture with better spatial coverage, shown in Fig. 14.





**Figure 14.** (Top) MODIS AOD averaged over the 2007-2016 period. (Middle) Average ModelE base case AOD, re-gridded through bilinear interpolation to match the 1º-by-1º resolution of MODIS data. Grid cells corresponding to those that do not have MODIS data are removed (shown in grey). (Bottom) Model AOD bias, calculated as the difference between ModelE base case and MODIS. Left column shows results for the month of March, while the right column shows results for the month of August, consistent with Fig. 13. BB regions of interest are outlined in black.

In the bottom row, we can see a positive model bias over regions heavily influenced by sea salt aerosols, like the Southern Ocean, as well as over Northern Africa, which is influenced primarily by dust. Such bias in dust and sea salt aerosols was not observed in previous CMIP6 model evaluation with MODIS comparison (Bauer et al., 2020), but the ModelE radiation scheme has since been updated with a more accurate treatment of aerosol hydration using Köhler theory, influencing aerosol wet radius and, therein, AOD. This suggests that the model AOD overestimation is different from previous results due to a change in model parameterization, not the BrC scheme presented here.



A strong positive bias can be seen over SEAS in March, which falls in the BB season of the region. Similarly, a slightly
weaker, though still prominent, positive bias can be seen over part of SHSA in August. Since this occurs during the BB
season of each respective region, it may indicate an overestimation of BB emissions. EQAS appears to be the only BB region
in which ModelE underestimates AOD: this can be seen as a slight negative bias over the region in Fig. 14 and is further
supported by Fig. A4. Since this bias appears stronger in August, during the peak of EQAS BB activity, this may be due to
an underestimation of emissions from peat burning, which dominates BB in the region (van der Werf et al., 2017). To
summarize, this analysis has afforded interesting insight into ModelE AOD biases, suggesting that the observed differences
between retrieved and simulated optical depth is largely a result of changes in model implementation rather than BrC
representation.

## 3.4 Study limitations

There are processes influencing BrC in the atmosphere that were not included in the work presented here, posing limitations
to our estimate of BrC radiative effect. Firstly, chemical aging of secondary BrC was not simulated, despite laboratory
studies showing secondary BrC undergoes bleaching. As mentioned in Sect. 2.2.4, our current BrC aging scheme does not
account for the semi-volatile nature of SOA, and therefore cannot be used to bleach secondary BrC. This may cause the SOA
contribution to BrC warming to be overestimated, as SOA absorption should decrease by at least 50% during the day, so we
plan to include this in future work. Aqueous phase aging of BrC, which has faster rates of browning and similar rates of
bleaching compared to heterogenous aging (Zhao et al., 2015; Hems et al., 2021), was also not included. Faster browning,
with limited change in bleaching, may increase BrC-induced warming. On the global, annual scale of analysis presented
here, however, it may not have a discernible effect, just as browning, which resulted in a build-up of more-absorbing BrC
overnight, showed no distinguishable effect in sensitivity tests.

While missing aging processes may cause an overestimation of the BrC radiative effect, there are some sources of BrC
that weren't introduced into ModelE, resulting in a possible underestimation of BrC absorption and, therein, radiative effect.
SOA originating from aromatic precursors have been shown to absorb light (Liu et al., 2016). ModelE, however, doesn't yet
have aromatic gases explicitly represented, and therefore doesn't have the ability to simulate aromatic SOA. Additionally, as
mentioned in the introduction to this study, recent work has identified a darker, more refractory, less soluble subset of BrC
closely resembling tar balls (Saleh et al., 2018). These aerosols have been shown to absorb not just in the UV-VIS
wavelength range, but also in the near-IR (Hoffer et al., 2017; Corbin et al., 2019; Chakrabarty et al., 2023). Since our
representation of BrC only accounts for absorption in the 300 to 770 nm range, this tar ball subset of BrC could constitute an
important source of organic warming in longer wavelengths. Further attention should be given to the sources and optical
properties of this subset, to allow for incorporation into climate modeling.



**4 Conclusions**

Carbonaceous aerosols like OA are expected to grow in importance as climate forcers, as wildfire frequency and intensity increase with climate change, yet OA forcing still contributes a large uncertainty ($\pm$ 0.23 W/m$^2$) to Earth system models (Flannigan et al., 2009; Keywood et al., 2013; Tsigaridis and Kanakidou, 2018; Szopa et al., 2021). To improve the physical and chemical correctness of OA, and allow for better calculation of OA forcing, light absorption of BrC aerosols must be accounted for in climate models. We presented the first implementation of BrC in the GISS ModelE ESM. BrC was

introduced to ModelE through the definition of four properties or processes: BB BrC-to-OA emissions ratio, attribution of absorption to biogenic SOA, imaginary RI of primary and secondary BrC, and a unique chemical aging scheme for primary BrC. We conducted sensitivity tests in which these properties were varied to, firstly, estimate the average radiative effect of BrC and, secondly, understand how that effect may change across a reasonable range of uncertain parameters. Finally, ModelE performance with BrC aerosols was evaluated by comparing simulated total AOD and AAOD to retrieval data from

AERONET and MODIS.

Both sensitivity tests and evaluation against retrieval data showed BrC has no discernible effect on total AOD and AAOD. There was no observable change in total AOD, AAOD, or therein SSA, between the control simulation with no BrC and all sensitivity test simulations. Further, comparison to retrieved optical depth showed similar model skill with and without BrC. Biases in ModelE AOD, namely an overestimation compared to retrieval AOD, were identified in this study,

but these were attributed to changes in model implementation, not the BrC scheme presented here. While BrC did not change model performance in terms of optical depth, it did reduce the total cooling effect of OA, contributing a net TOA radiative effect of 0.04 $\pm$ 0.01 W m$^{-2}$, based on the global annual average of our base case simulation. Therefore, the physical and chemical complexity introduced by BrC may not be necessary to improve ModelE AOD or AAOD performance, but it should be included to increase the accuracy of OA radiative forcing estimates.

With regards to BrC parameters that were represented in this study, sensitivity tests showed that separating BrC from other organics, including secondary BrC, and simulating chemical bleaching all had distinguishable radiative effects and, as these properties are consistent with laboratory studies, should be accounted for. Variation in primary BrC imaginary RI and BrC-to-OA emission ratio, as well as simulation of chemical browning, did not show distinguishable effects. This indicated that in the scope of global, annual average radiative effect, it is not critical to precisely define values for these properties and

browning can be ignored. On smaller spatial and temporal scales, however, these may be of greater importance. Since model evaluation with total AOD and AAOD provided no insight into BrC properties, our next step is to further constrain this parameter space.

There have been in-situ measurements of BrC absorption, in or downwind of fires, measured during flight campaigns (Zhang et al., 2017; Zeng et al., 2020, 2021; Washenfelder et al., 2022; Zeng et al., 2022) as well as retrievals of BrC

properties, mass and optical depth, from the AERONET and IMPROVE ground-based networks (Arola et al., 2011, 2015; Schuster et al., 2016; Chow et al., 2018) and satellite data (Li et al., 2020b, 2022). These can be used to directly evaluate



ModelE BrC representation. In a future study, which is already underway, we will present model evaluation against such in-situ measurements of BrC absorption in addition to a retrieval of BrC AOD and AAOD from AERONET (Schuster et al., 2016). By comparing these BrC specific data to that of ModelE, we hope to constrain the BrC parameter space defined here, 695 and further improve OA representation.

## 5. Recommendations for complexity of BrC representation

Based on the findings and conclusions of this study, we make recommendations on the degree of complexity needed for BrC representation within an Earth system model like ModelE, with the caveat that this is model dependent. Our recommendations are inherently dependent on the research objective of simulating BrC aerosols.

If the objective is to capture total AOD and AAOD at 550 nm, the irradiance weighted average wavelength in the solar spectrum, no explicit BrC representation is needed. However, the same cannot be said for capturing total optical depth at shorter, UV/near-UV wavelengths, where BrC absorption is maximized and likely contributes more to total AAOD. If the research objective is to estimate global, annual average TOA BrC radiative effect, BrC should be explicitly represented, biogenic SOA should be treated as brown (though much less absorbing than primary BrC), and a BrC bleaching process 705 should be simulated. In this case, the imaginary refractive index of primary BrC at 550 nm and the BrC-to-OA emissions ratio do not need to be strictly defined. We instead recommend applying a reasonable range for these parameters: 0.003-0.03 for $k_{BrC,550\,nm}$ and 15-55% for BrC-to-OA BB emissions proportion (see Sects. 2.2.1 and 2.2.3). Regarding regional studies of BrC radiative effect: though our regional sensitivity analysis yielded no additional conclusions, parameterizations for $k_{BrC,\,550\,nm}$ and BrC-to-OA emissions ratio can be tailored to specific BB regions given prior knowledge of regional BC and OA 710 emissions. This would allow for a region-specific, likely narrower, range of parameters to be utilized. Finally, if the research objective is to capture the diurnal variability of OA absorption, a browning process should be included in addition to the processes and parameters for the global radiative effect case. Both browning and bleaching processes should be linked to and driven by hydroxyl and nitrate oxidant, as well as ozone, concentrations, to allow for the build-up of more absorbing BrC at night via nitrate oxidation and more rapid aging during the day.

*Code and data availability.* The GISS ModelE code is publicly available at https://simplex.giss.nasa.gov/snapshots/; the most recent public version is E2.1.2. The Fortran code used for the simulations described in this study, along with model output and Alaskan peat sample input and fitted data (see discussion on "first parameterization" in Sect. 2.2.3), is available here: https://doi.org/10.5281/zenodo.8342620. Model code can be found in the file titled "modelE_code_092723.tar.gz", model output is in the file titled "ModelESimAndEmisData.tar.gz", and Alaskan peat data is in the excel file titled "KK 720 Parameterization-AK Peat.xlsx". Model simulation data are averaged over specified time periods and included as netCDF files; individual file names start with the period averaged over and end with the simulation type ("SensitivitySim{#}" or "transient_{ctrl/base}case"). The CEDS emissions file used for equation 1 (see Sect. 2.2.1), titled



"CMIP6_CEDS_BBURNemis_forEq1.nc" is also included with this simulation data. MERRA-2 reanalysis data are available at https://gmao.gsfc.nasa.gov/reanalysis/MERRA-2/. AERONET data are available at https://aeronet.gsfc.nasa.gov/. Lastly,
MODIS data are available at https://modis.gsfc.nasa.gov/.

*Author contributions.* MAD, KT, and SEB conceived the study. All model development was done by MAD, guided by KT and SEB. JC fit the Alaskan peat data to inform BrC RI and drafted the language describing the parameterization (see Sect. 2.2.3). GLS provided guidance on the use of AERONET retrieval data for ModelE comparison. MAD conducted all model simulations and analysis, created all figures, and drafted the first version of this manuscript. All authors contributed to later
drafts.

*Competing interests.*  Kostas Tsigaridis is a member of the editorial board of Atmospheric Chemistry and Physics.

*Acknowledgements.* Climate modeling at GISS is supported by the NASA Modeling, Analysis and Prediction program. MAD acknowledges support from the Future Investigators in NASA Earth and Space Science and Technology program
(grant number 80NSSC22K1442). KT and JC acknowledge support from the Plankton, Aerosol, Cloud, ocean Ecosystem project (grant number 80NSSC20M0205). MAD and JC acknowledge helpful discussions with Hans Moosmüller on the complex refractive index spectra of BrC in atmospheric combustion aerosols. MAD acknowledges useful guidance from Róisín Commane and Faye McNeill. We thank the principal investigators (PIs) of the AERONET network and their staff for establishing and maintaining the different sites used in this investigation. The Terra/MODIS Aerosol L2 dataset was
acquired from the Level-1 and Atmosphere Archive & Distribution System (LAADS) Distributed Active Archive Center (DAAC), located in the Goddard Space Flight Center in Greenbelt, Maryland. We thank the PIs, Rob Levy and Christina Hsu, for making available their aerosol dataset. Resources supporting this work were provided by the NASA High-End Computing (HEC) Program through the NASA Center for Climate Simulation (NCCS) at Goddard Space Flight Center.



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



**Appendix A:**

| Aging process | Oxidant | Second order rate constant |
|---|---|---|
| Browning | OH | 1.9e-11 cm$^3$ molecule$^{-1}$ s$^{-1}$ |
| | NO$_3$ | 1.7e-13 cm$^3$ molecule$^{-1}$ s$^{-1}$ |
| Bleaching | OH | 4.4e-11 cm$^3$ molecule$^{-1}$ s$^{-1}$ |
| | O$_3$ | 9.15e-16 cm$^3$ molecule$^{-1}$ s$^{-1}$ |

**Table A1.** ModelE prescribed second order rate constant for each BrC aging reaction driven by atmospheric oxidants. Constants are derived from a kinetic model provided by Hems et al. (2021).



| Biomass burning region | Months considered peak biomass burning period |
|---|---|
| Southern Hemisphere South America (SHSA) | August, September, October |
| Southern Hemisphere Africa (SHAF) | July, August, September |
| Temperate North America (TENA) | June, July, August |
| Boreal North America (BONA) | June, July, August |
| Southeast Asia (SEAS) | March, April, May |
| Boreal Asia (BOAS) | March, April, May |
| Equatorial Asia (EQAS) | July, August, September |
| Australia (AUST) | September, October, November |

**Table A2.** Months considered peak fire period for each biomass burning region in model evaluation against AERONET and MODIS data. These months are based on periods of peak emission as discussed in Pan et al. (2020).

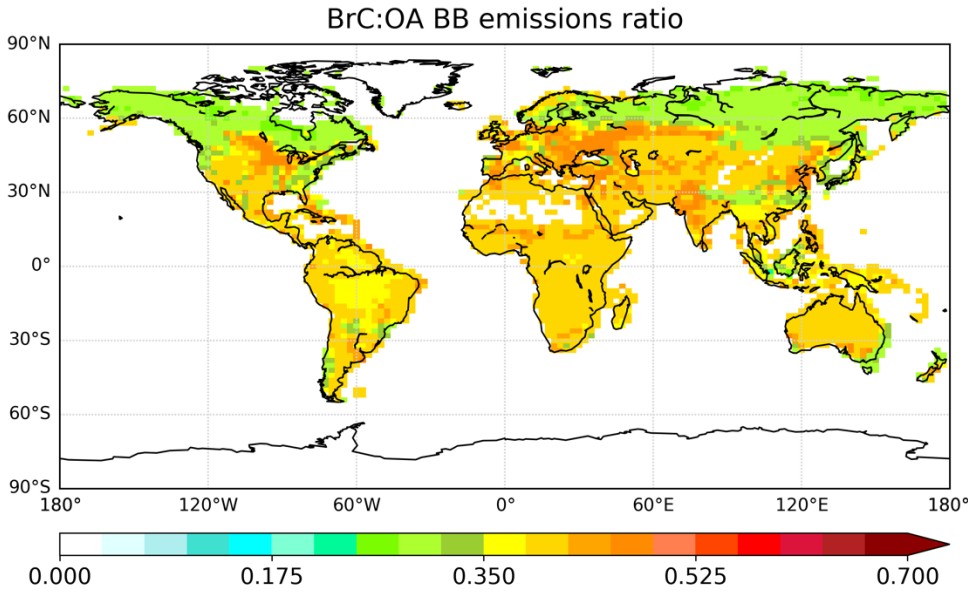

**Figure A1.** BrC-to-OA emissions ratio, calculated according to equations 1 and 2 using BC and OA emissions from the CEDS BB inventory (year 2000 climatological monthly emissions averaged over the entire year). White space shows where BC or OA emissions are zero.

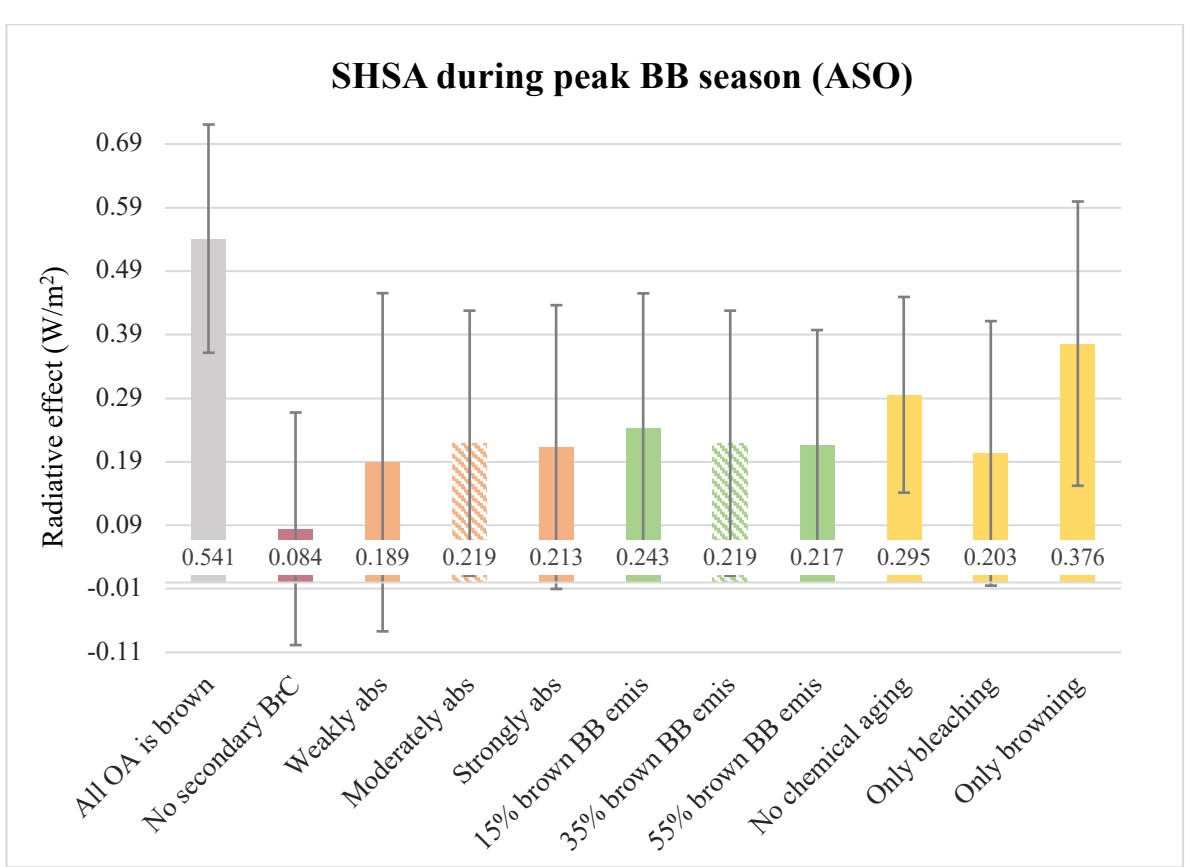

**Figure A2.** Radiative effect of each BrC simulation, averaged within the Southern Hemisphere South America (SHSA) BB region across months of peak fire activity–August, September, and October (ASO; see Table A2). Consistent with Fig. 9, BrC effect is calculated according to equation 8. Error bars show the standard deviations (variability of each simulation across repeated years of simulation), different colored bars indicate a different BrC property varied, and dashed bars indicate the base case of BrC representation (shown twice for ease of comparison to other simulations), consistent with Figs. 7, 9, and 10.



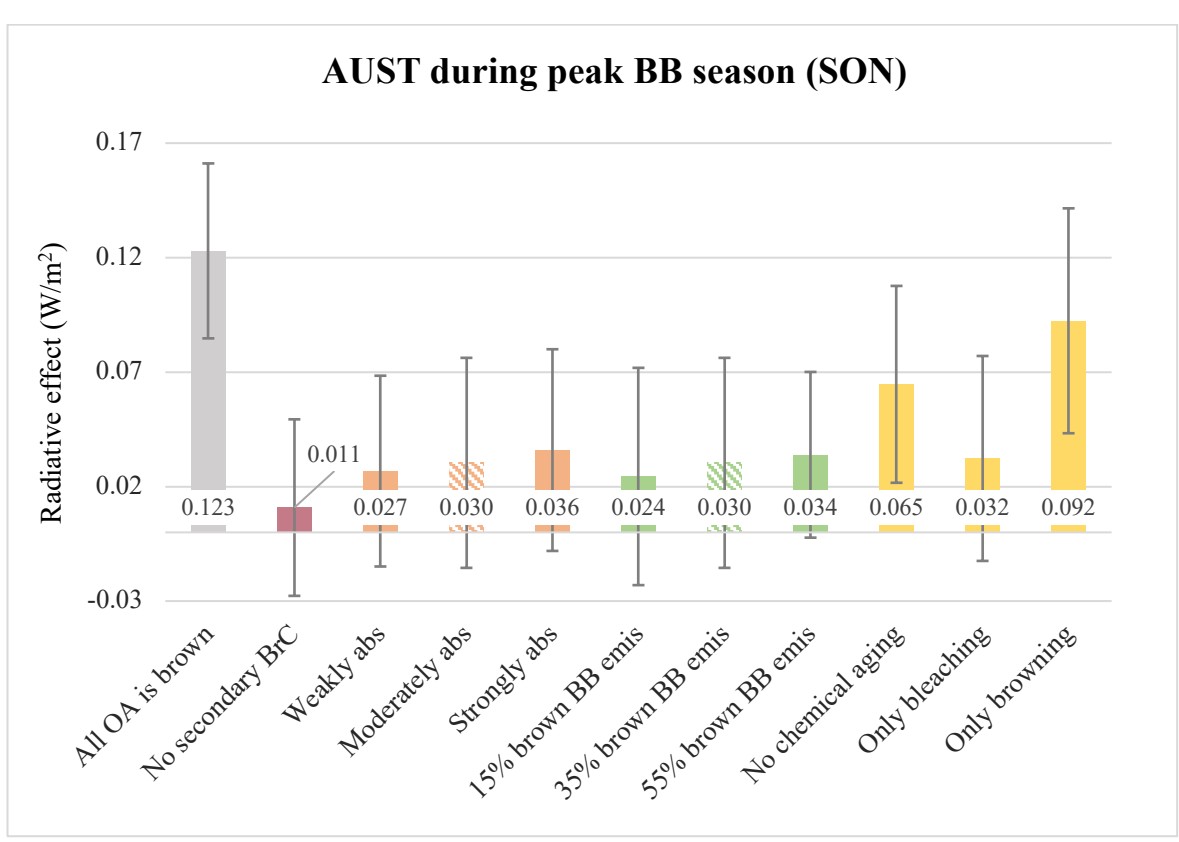

**Figure A3.** Radiative effect of each BrC simulation, averaged within the Australia (AUST) BB region across months of peak fire activity–September, October, and November (SON; see Table A2). BrC effect is calculated according to Equation 8, and displayed error bars, bar color, and dashed bars are consistent with Figs. 7, 9, 10, and A2.





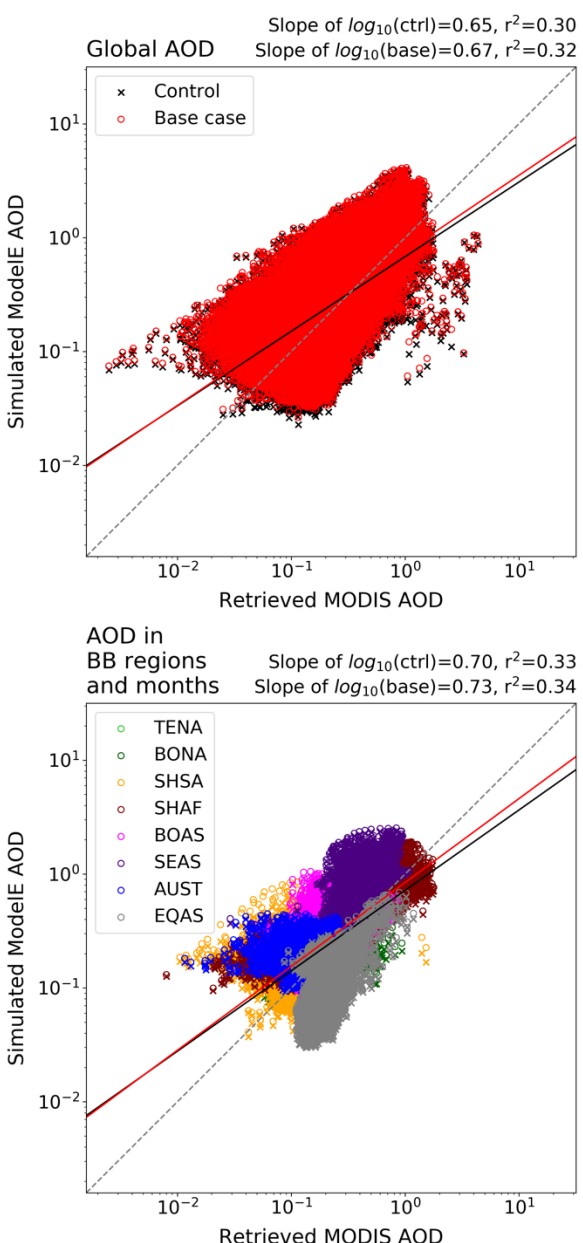

**Figure A4.** Simulated ModelE optical depth at 550 nm plotted in the $\log_{10}$ space against retrieved MODIS optical depth. Each point corresponds to the optical depth in one month, averaged across the 2007-2016 period, in each grid cell (ModelE has been re-gridded to match MODIS' 1º by 1º resolution). ModelE control case optical depth values are shown as 'x's, while base case values are shown as 'o's. The slope and $r^2$ of the linear regression are displayed on the top-right corner of each plot, and regression lines are included within each plot. (Top) Global AOD values. (Bottom) AOD in BB regions during months considered peak for BB, with each color representing a different region.