# Peer review of "Modeling atmospheric brown carbon in the GISS ModelE Earth system model"

_EGUsphere, 2023_

## Author Comment (AC1)

We thank both reviewers for the helpful revisions that have improved the manuscript. This document includes the comments of both reviewers as well as our responses. Reviewer comments are **bolded**, responses are in regular font, and *excerpts of changes to manuscript are in italics, with new changes underlined if added to an existing sentence. We differentiate*

5 reviewer 1 and 2 comments with underlined headings. We also include references cited in responses at the end of this document.

**Comments from anonymous referee #1:**

This work introduced BrC to the One-Moment Aerosol module of the GISS ModelE Earth system model. The different part from previous similar modeling studies might be the implementation of BrC from BVOCs and the Hems et al. aging. The manuscript is in general well written. However, I have some main concerns for the authors to consider:

- 1. It seems that all BrC in the model is assumed to be water-soluble, is it true? Many references that cited in the manuscript have suggested that water-soluble BrC might 15 only contribute ~half of the absorption. The solubility of BrC, or the fraction that dissolves in water, is set as 0.8 (see submitted Table 1). While this is higher than some literature estimates (Laskin et al., 2015), it is within reported ranges; in studying BrC solubility and absorption, Zhang et al. (2013) found up to 82% of organic carbon (OC) filter extracts were water soluble (this citation has been added to submitted L270). We 20 use this higher solubility fraction because it is consistent with ModelE default organic aerosol (OA) tracers. If the solubility of primary BrC (the only tracer we added to ModelE) were decreased, for instance to 0.5 for emitted BrC, the primary effect is an increase in BrC and, therefore, total OA burden. In a trial sensitivity test of this, we found such a change increased total organic burden by 26%. Such an increase in organic mass 25 results in a net cooling effect for BrC of  $-0.16\pm0.16$  W m-2: with all other parameters being the same as the BrC base case, this simulation has more organic mass, resulting in even more scattering compared to the control simulation. We summarize this by adding text to the end of section 2.2.3 (submitted L272): "These physical properties were kept 30 constant to maintain consistency with ModelE default OA representation and ensure no change in total organic mass burden with the introduction of the BrC scheme. The solubility of BrC, for example, is left at 0.8, which is higher than some literature estimates (Laskin et al., 2015) but within reported ranges (Zhang et al., 2013), because a lower value would result in an increase in BrC, and therefore total OA, burden. An increase in organic mass, compared to the model default, would result in an increase in 35 scattering and a substantial cooling effect, negating the purpose of the BrC scheme, which is to account for OA absorption and the subsequent warming effect. Thus, we changed only organic optical properties to represent BrC-total organic mass burden was not changed."
- 40 It's also worth noting that the fraction of BrC mass that is water soluble (WS) is not the same as the WS contribution to absorption. We do not differentiate WS-BrC from water-insoluble BrC, and therefore do not assign differing optical properties, to, again, be consistent with the model: BB OA and biogenic SOA, the two default tracers that become BrC, are not differentiated by solubility or hygroscopicity in ModelE.

45 2. The comparison of AOD/AAOD to measurements is not very helpful. Maybe it is because the author said extensive work of evaluation will be presented in a future study. The authors focused on AOD at 550nm which is not very relevant to BrC. Alternatively, I think comparison of AAOD to AERONET can be elaborated, e.g., considering AAOD at different wavelengths, which could give much more useful information. The primary purpose of comparison of AOD/AAOD to measurements was 50 to assess general model ability to capture total aerosol optical properties-essentially, does the BrC scheme improve or impair overall model performance. For this reason, we looked at total aerosol optical depth and worked within the current confines of ModelE radiation, which is output in broad wavelength bands (see submitted L114). We provided discussion on the potential limitations of using a broad wavelength band, with optical 55 depth indicative of 550 nm (see submitted L200-210 and L586-592). Looking at AAOD at shorter wavelengths, as suggested by the reviewer, would require applying an assumed absorption Ångström exponent (AAE) to ModelE BrC optical depth, acting as a scaling factor, introducing further uncertainty to the analysis. We chose to focus on total 60 AOD/AAOD performance, rather than specific to BrC at shorter wavelengths, because, as pointed out by the reviewer, we are currently carrying out a study that will do this extensively. This was summarized with the addition of text to submitted L407: "We focused on optical depth at 550 nm, rather than a shorter wavelength, because the purpose of this evaluation was to assess general model ability to capture total aerosol 65 optical properties; essentially seeing if the BrC scheme improves or impairs overall model performance. For this reason, we worked within the current confines of ModelE radiation, which produces output in broad wavelength bands, as mentioned in sections 2.1 and 2.2.3, indicative of 550 nm in the UV-VIS band. Analysis at a shorter wavelength would require assuming an Ångström exponent for ModelE BrC optical depth, 70 introducing further uncertainty to the parameter space. We will do such analysis in a future study, where BrC absorption will be evaluated extensively."

To expand on the overview given in submitted L688-695: we are comparing ModelE BrC optical depth and mass to a BrC-specific retrieval from AERONET (Schuster et al., 2016), as well as BrC absorption to in-situ measurements from the DC3 (Zhang et al., 2017), SEAC4RS (Zhang et al., 2020), ATom (Zeng et al., 2020), FIREX-AQ (Zeng et al., 2021), and WE-CAN (Sullivan et al., 2022) flight campaigns. For these comparisons, we are specifically seeing if this BrC scheme captures daily and sub-daily BrC absorption. We also added the following language to submitted L694 to address this concern more directly: "*We perform these comparisons either at 550 nm, when provided by the data, or at more BrC relevant wavelengths by applying AAE suggested by campaign PIs to ModelE output.* By comparing these BrC specific data to that of ModelE, we hope to constrain the BrC parameter space defined here, specifically evaluate *performance of BrC absorption, and further improve OA representation.*"

75

80

 3. The authors provided their recommendations for BrC model representation in Section 5, but did not explain clearly how these recommendations are made. Due to the lack of model evaluation, it is hard to point out which mechanisms/sensitivity simulations in this work are more likely to be correct than others. Since model evaluation against AERONET and MODIS data provided no constraint on the BrC

2

scheme, these "recommendations" are based on our best estimates of parameters,
discussed in Section 2.2, and the subsequent sensitivity tests of these parameters,
presented in submitted Section 3.2.2. We acknowledge, however, that the term
"recommendations", when based on this kind of analysis, may be misleading. In response to this, we've reframed Section 5 with a clearer purpose: to summarize the development portion of this study, we present best practices for modeling BrC in ModelE to maximize
accuracy while minimizing computational cost (edits made across Section 5, starting on submitted L696). The remainder of this response (as well as our response to specific comment #11), discusses exactly how we arrive at these "best practices" for ModelE.

As pointed out by this comment, sensitivity tests cannot discern which change in a parameter is correct, just the effect it has on BrC radiative effect. When the BrC effect 100 between two simulations is distinguishable, we take the parameterization that is supported by our literature analysis to be correct. For instance, the simulation in which secondary organic aerosols (SOA) are not brown has a distinguishable effect from our base case (where they are); since previous studies suggest SOA is brown, we identify that as more accurate for BrC representation. Another example is chemical aging: simulations 105 with no chemical aging or just browning have distinguishably higher radiative effects than the base case, and we know BrC bleaches in the atmosphere, so the base case is considered more correct. When we cannot distinguish simulations by their radiative effect, we look at the scale and objective of analysis to determine the best practice for ModelE (e.g., only bleaching is enough to represent average aging over long time scale, 110 removing the computational cost of an additional BrC tracer). Finally, if we can't distinguish parameters by scale or research objective, we leave parameters in their ranges computed in Section 2.2 (see response to specific comment #11).

**Specific comments:**

120

115

p.6, unit of kgC is used for BrC and OA emissions for BrC/OA ratio. Could you please also indicate whether kgC or kg was used when deriving BC/OA ratio in equation (1) and (5)? If you used kg for OA emission here, what is the OC/OA ratio in your calculation? Since each of these equations uses emissions from the CEDS inventory, units for all of them are in kgC. This has been clarified in submitted L167 ("...EOA and EBC are OA and BC emission in kgC...") and submitted L241 ("...a function of the BC-to-OA emissions, in kgC, ratio from the CEDS inventory..."). Generally, ModelE input files for carbonaceous aerosols provide emissions in kgC, while ModelE output is in kg, with the model using a constant OC/OA ratio of 1.4 (Tsigaridis et al., 2014).

p.7, line 192. I think this statement could be clearer. Aromatic SOA may contribute small to OA mass, but its absorption efficiency could be larger than biogenic SOA. I think talking about secondary BrC mass is meaningless as it is highly sensitive to your assumption that non-zero imaginary RI means 100% BrC. This statement has been edited for clarity: "Aromatic SOA are not yet represented in ModelE, since they are small contributors to the global OA budget (Tsigaridis and Kanakidou, 2003). Despite their smaller burden, aromatic SOA are typically more absorbing than biogenic SOA (Liu et al., 2016), creating a potential low bias in secondary BrC absorption."

3. Table 1. I do not think one single number for the solubility and k is appropriate. According to Laskin et al., 2015 that cited here, water soluble and insoluble BrC could contribute similar mass. Even for BC and OC, models usually treat their wet scavenging based on their solubility (e.g., fresh vs aged). See response to main concern #1.

135

160

165

170

175

4. p.10, line 277. The two numbers, 150% and 20%, can bring huge uncertainties and need more discussion. In addition, it seems that the reference that cited, Zhao et al., focused on water-soluble BrC only. This is a great point: these relative absorptions, like 140 the refractive index of emitted BrC, are based in literature but inherently uncertain. The 20% threshold absorption is based on laboratory studies, including Fleming et al., (2020) in which pine burn chromophores' absorption was observed to decrease between 70-90% with bleaching as well as relative absorption decrease due to OH and O3 heterogenous oxidation shown in Figure 14 of Hems et al. (2021). Additionally, our assumed value is 145 close to the threshold used in previous BrC modeling studies (Wang et al., 2018). The upper limit of 150% is more uncertain, as there's fewer studies of this. We agree with the reviewer that Zhao et al. (2015) is not the proper citation here as this refers to WS-BrC aging. Zhong and Jang (2014) observed an increase of 11-54% absorption through browning, while Hems et al. (2021) suggested a much higher increase due to nitrate oxidation (4 times initial absorption). Thus, 150% relative absorption is near the middle 150 of this literature range. New language has been added to submitted L278-280 to include these references and remove Zhao et al.: "The threshold value of 20% relative absorption was based on laboratory studies of oxidized BrC proxies (Fleming et al., 2020; Hems et al., 2021) and is close to the threshold value used in other modeling studies (Wang et al., 2018). 150% relative absorption is used as it's near the middle of the range of reported 155 photo enhancement in laboratory studies (Zhong and Jang, 2014; Hems et al., 2021)."

We agree with the reviewer that these uncertain parameters warrant further discussion.
As such, we've added discussion points within the manuscript. To submitted L347: "We should note that while the relative absorptions of aged BrC tracers are also uncertain parameters, we did not vary these in sensitivity tests, focusing first on the impact of simply including or excluding aging processes." To submitted L538: "...simulating browning appears unnecessary on the scale of global annual averages, if the only interest is BrC radiative effect. This further suggests refining the relative absorption value of browned BrC, now 150%, may not be necessary for this scope of study." Finally, to submitted L682: "Because bleaching has been identified as a key process, the effect of varying the threshold absorption of primary BrC should be investigated in future work."

5. Section 2.2.4, what about the mass change due to the classic OA aging? It would affect the mass and density so also affect MAE and absorption. The "classic" OA aging represented in ModelE changes hygroscopicity of anthropogenic OA, not BB OA, going from hydrophobic to hydrophilic (Koch, 2001). This aging for anthropogenic OA does not affect aerosol mass and density: OA mass moves between tracers or is removed via sinks, and density remains constant. While a change in density may occur over time in the atmosphere and affect MAE, we do not simulate this to remain consistent with the default model approach. As mentioned in the responses to main concern #1, when simulating BrC, altering total OA mass in the atmosphere, rather than just shifting mass

between tracers/altering optical properties, can result in large changes in organic radiative forcing which would mistakenly be attributed to BrC.

- 6. Fig 6,7,9,10-13, could you please add wavelength information to the figure title or captain? In submitted Figures 6 and 7 captions, we have clarified "...SSA, in the UV-VIS band". The wavelength range of the UV-VIS band was previously defined in 180 submitted L199-200. The caption for submitted Figure 9 already specifies that radiative effect of BrC was calculated according to Equation 8 (submitted L514), which uses radiative forcing, a sum of shortwave and longwave forcings (see submitted Table 3 caption). For submitted Figure 10 and submitted Figure 12 captions, this was clarified as 185 "UV-VIS AOD" and/or "UV-VIS AAOD". The caption for submitted Figure 11 already specifies the wavelength (see submitted L558), but explicit mention of the UV-VIS band was added to help clarify this: "Simulated ModelE optical depth at 550 nm (spectrally weighted average of the UV-VIS band) plotted ... ". For submitted Figure 13 caption, this was clarified as "...corresponds to the different in ModelE base case and AERONET optical depth, at 550 nm, at an AERONET site ... ". Though it was not mentioned, we 190 added a similar clarification to the caption of submitted Figure 14: "MODIS AOD at 550 nm...".
- 7. I am confused that throughout the manuscript, while BrC radiative effect is seemed to be focused (e.g., in eq 8), radiative forcing is also frequently used later. Please make this clearer. Throughout the submitted version, any mention of "forcing" refers to the TOA radiative forcing (RF) of an aerosol and is used only for total organics or other aerosol species. Whenever discussing BrC, the term "effect" is used, calculated according to Equation 8. Submitted L541 mistakenly referred to the BrC effect as "forcing", which we have corrected.
- 8. p.19, paragraph 2, how are those +/- 0.1 uncertainties calculated? These are better referred to as "variabilities" and are calculated as the standard deviation of a particular simulation. These are calculated the same way as the error bars in submitted Figures 9, 10, A3, and A4 are (see explanation in each respective submitted caption). To clarify this, an additional sentence was added at the end of submitted L459: "Variabilities presented here are calculated as the standard deviation across repeated years of each simulated case."
  - **9.** Table 3, I am very surprised that longwave RE of dust is larger than its shortwave RE. After reviewing this, we agree the magnitude of longwave dust radiative forcing is higher than expected. However, dust radiative forcing is outside the scope of this study, as the BrC scheme is separate from dust aerosols; it was initially included to simply serve as a reference. To avoid confusion, we refine the reference radiative forcings in submitted Table 3, limiting it to just accumulation mode aerosols, changing submitted L463-464 as follows: *"For reference, Table 3 shows the TOA, instantaneous direct RF of other ModelE simulated accumulation mode aerosols."* Following this, we also remove discussion of dust radiative forcing from submitted L467-468.

210

215

10. The structure of Section 3 looks confusing. I suggest comparing AOD etc. with measurements first and then discussing radiative forcing. We agree and have restructured the results section. Section 3.1 now looks at changes in ModelE absorption and AOD/AAOD across sensitivity tests, section 3.2 details the comparison with measurements, and section 3.3 discusses radiative effect. Figure numbers have been adjusted, as a result.

11. p.706-707, how are these suggested ranges derived? As there is no observational constraint in this work, how a "reasonable range" could be recommended? Following from our response to main concern #3, ranges of refractive index and BrC-to-OA emissions ratio are suggested because sensitivity tests are unable to distinguish the radiative effects between simulations using different range values. In other words, we couldn't narrow these ranges, so, for now, using a value within the range is most accurate. These ranges, 15-55% for BrC-to-OA emissions proportion and 0.003-0.03 for  $k_{BrC, 550 \text{ nm}}$ , are derived in Section 2.2.1 and 2.2.3, respectively.

**230 Comments from anonymous reviewer #2:**

The study presents a new methodology to introduce brown carbon aerosol absorption in the GISS ModelE Earth system model. This includes browned (stronger absorbing) and bleached (weaker absorbing) organic aerosol tracers that are transformed from the primary emitted tracer depending upon the concentrations of oxidants in the atmosphere.

- Several sensitivity runs are performed to understand the effect of the different parameters required for modelling BrC optics in the model. Overall, the manuscript is, in general, clearly written and presents an important improved framework to previous treatment of brown carbon absorption in climate models that aids in increased accuracy of their radiative effects. However, some changes are suggested below, that the authors may consider to improve the manuscript
- 240 consider to improve the manuscript.

**Major comments:**

225

1) The one moment aerosol (OMA) module is used to calculate the aerosol optical properties in the model. However, a description of how this module makes optical property calculations is missing from the manuscript, which makes it difficult to understand how the

245 brown carbon aerosols were incorporated. A brief description of the module, including the aerosol mixing scheme is suggested to precede the discussion on how BrC was added to the module.

A more thorough description of OMA radiation module was added to submitted L115. The edited text is as follows: "To account for aerosol swelling with water vapor, dry aerosol size,

- 250 relative humidity, aerosol hygroscopicity and the refractive index of water are used, with Köhler theory as a base for calculation, to obtain wet aerosol radius and complex refractive index. Apart from swelling with water, there is no internal mixing in OMA radiative calculations–all aerosols are considered externally mixed. Wet aerosol size, as well as real and imaginary refractive index are then used to find corresponding aerosol scattering, asymmetry, and light
- 255 extinction values in pre-calculated Mie look-up tables. These optical properties, computed for six wavelength bands in the shortwave (SW) and 33 in the longwave (LW), are used to calculate ARI (Bauer et al., 2010)."

2) Lines 98-99: Wang et al. (2018) have also simulated chemical aging based on OH
 concentrations. So, the present study may not be the first attempt to incorporate a concentration-dependent aging scheme. Also, it would be interesting to see if such oxidant concentration-dependent aging schemes perform any differently from fixed-time aging schemes. Yes, Wang et al. (2018) simulated aging by changing BrC optical properties over an e-folding time partially determined by hydroxyl concentration. We acknowledge their use of

- 265 hydroxyl concentration in aging in submitted L325. The difference between our aging scheme and theirs, beyond the fact that we incorporate browning while they do not, is that they change optical properties of one BrC tracer over time, while we move mass between tracers with prescribed optical properties. Our chemical aging scheme is the first to move mass between tracers with different optical properties based on oxidant concentrations (Drugé et al., 2022
- 270 moved mass between tracers using a set lifetime–"fixed-time aging", as this comment calls it).

We agree with the reviewer that is would be interesting to see how a fixed-time aging scheme compares to an oxidant-driven scheme, particularly to see if there is a difference that would make the added complexity necessary. We ran a quick simulation to try to investigate this: we set our aging scheme to transfer mass according to a fixed time rather than second order rate constants (though that fixed time is still based on Hems et al.'s kinetic model). The fixed-aging case resulted in a global average radiative effect 0.01 W/m-2 less than the base case–a small difference considering the variability of base case is the same magnitude. However, fixed aging does result in differences in global distribution of the BrC radiative effect:

280

285

290

Understanding these differences is not trivial as both BrC and atmospheric oxidants, particularly hydroxyl, have complex profiles. Despite the small global average difference in radiative effect, we do see some larger magnitude differences between cases occur over our BB regions of interest. Such observed differences in this quick check, and the fact that oxidant-driven chemistry allows for more accuracy, reaffirm our choice to use an oxidant-driven scheme.

3) OA has been used throughout the manuscript to denote organic aerosols with units as Tg C /yr in some places (eg. line 174) and just Tg /yr in others (eg line 139). This is confusing as OA is generally used as the total organic aerosol mass and OC only the carbon mass in OA. I suggest that this distinction be made clear when it is first mentioned, and the terms

used appropriately in the manuscript. This point of confusion was similarly noted by the first

275

reviewer (see review #1, specific comment #1). In addition to our response to this earlier comment, we have added clarifying language to submitted L169, prior to use of "kgC" in submitted L174: *"It's important to note that emissions inputs for organic aerosols are in units of*

- 295 mass of carbon, while ModelE output, and most of this discussion, uses total organic aerosol mass. To convert between these, ModelE uses an organic carbon (OC) to OA ratio of 1:1.4 (Tsigaridis et al., 2014)." We have also added clarifying language to submitted L478, preceding a paragraph where we use both "Tg" and "TgC" to compare mass and emissions of different models: "To compare our scheme with literature values that report organic mass in TgC, we
- 300 converted ModelE mass and emission output, in Tg OA, using the previously mentioned OC to OA ratio of 1.4."

4) dark-BrC particles exhibit BC-like optical properties, but emission inventories likely identify them as OC emissions as they are thermo-optically defined. Meaning that neglecting them in the present simulations may not be insignificant. There is evidence that

these particles are relatively resistant to photobleaching and possibly significant considering that the absence of photo-bleaching increases the BrC radiative effect, as demonstrated in the manuscript. Could these particles be incorporated into the present framework? As mentioned in this comment, the preliminary study of these aerosols did find

- 310 they resist daytime bleaching while still demonstrating nighttime browning (Chakrabarty et al., 2023). Thus, they could have a significant radiative effect. Unfortunately, it's too early in the study of dark BrC to include them in the present framework. Firstly, since dark BrC has only been observed in one published field study, we are severely limited in our knowledge of global sources, and therefore have no way of knowing how to include their emissions as a portion of
- 315 organics. Secondly, as we noted in submitted L63-65, current characterization of the optical properties of these aerosols makes them nearly indistinguishable from BC. We expanded our discussion of this on submitted L63-65, based on this answer: "Because there is limited observation and characterization of these aerosols, we have no way of knowing how to include their emissions as a portion of organics in ModelE. Additionally, initial work by Chakrabarty et
- 320 al. (2023) suggests its single-scattering albedo (SSA) and absorption Ångström exponent (AAE) are indistinguishable from that of BC. Given we have no knowledge of how to treat these aerosols, beyond the same as BC, we did not explicitly represent this subset of BrC."

Considering these and other limitations already highlighted, some statements in the manuscript regarding the insignificance of the optical properties of BrC and BrC-to-OA emissions ratios (eg. Line 539 and 614) may need to be revisited. Especially when considering the uncertainty associated with the optical properties of brown carbon particles and their variation with aging.

We want to clarify in response to this that BrC imaginary refractive indices and BrC-to-OA
emissions ratios are not referred to as "insignificant". We state that, because there was no
distinction when varying them between sensitivity tests, they don't need to be precisely defined
to study global average BrC radiative effect. This is further clarified in Section 5, where we state
a reasonable range of these values must be used (see response to review #1, specific comment
**11).**

335

305

**Specific comments:**

- Line 27: "aerosol produced from fuel and biomass burning..." - what kind of fuel? Here, "fuel" refers to both fossil fuels and biomass fuels, or biofuels. Within emissions inventories, fossil fuels are primarily coal, oil, and natural gas (Hoesly et al., 2018). Biomass fuel, or biofuel,

- 340 typically has a higher oxygen content then the hydrocarbons that make up fossil fuel, and makes use of waster or plant matter to produce energy (Demirbas, 2008). Since "fuel" encompasses both fossil and biofuel, we use this term for simplicity.
   Line 40: "SOA from biogenic VOCs (BVOCs) are also expected to grow in importance" -
- why? SOA are expected to grow in importance because anthropogenic emissions of other
  aerosols have been and will continue to decrease because of emissions controls, for instance the
  U.S. Clean Air Act in which particulate matter is regulated as a "criteria pollutant" (Schmalensee and Stavins, 2019), and cleaner technologies, as mentioned in submitted L37. We edited submitted L37 to mention emissions controls: "...and emission controls and cleaner technologies possibly lead to a further reduction of other aerosol sources (Bauer et al., 2022),
- 350 carbonaceous aerosols including OA could possibly become more prominent."
   Line 43: Mention which assessment report of IPCC Clarified in submitted L43: "...IPCC Sixth Assessment Report..."

- Line 50: "... incomplete combustion and smoldering fires ..." - smoldering fires also have incomplete combustion, please rephrase. Rephrased as: "It is emitted by smoldering fires and other incomplete combustion."

- 355 other incomplete combustion."
   Line 121: "Carbonaceous aerosols include BC and OA, which are each separated into aerosols from industrial and BB sources" what about other sources like transport and energy production? "Industrial" sources can be better understood as anthropogenic sources. To clarify this, we've changed "industrial" to "non-BB anthropogenic" on submitted L121. Sources
- 360 like transport and energy fall under this category in ModelE. Other anthropogenic sectors included are industry, solvents, shipping, agriculture, agricultural waste burning, and residential/other. We make the distinction of "non-BB" because some BB can be anthropogenic in nature, and we do not want to suggest otherwise.

- Line 133: Where are the biomass and industrial plumes emitted in the transient

- 365 **simulations and how are these expected to alter aerosol lifetimes?** Since plume injection heights are prescribed by GFAS1.2, which is a BB emission inventory (see submitted L131-132), this only changes the vertical injection height of biomass plumes. It has no effect on industrial plumes. Where biomass plumes are emitted differs between grid boxes with each fire. We have added the following clarifying language to submitted L133: "GFAS1.2 was used, rather than
- 370 *other fire emissions inventories, as it allows for implementation of plume injection height in each grid cell..." With regards to aerosol lifetime: we only ran transient simulations with GFAS1.2 emissions, so we can't comment on comparative aerosol lifetime as we don't have alternative lifetimes for analysis.*

- Line 166: MAC\_BrC\_550 is kept at a fixed value (1  $m^2/g$ ) but this value changes with

- 375 **changing k\_BrC (see Saleh, 2020). This may bias the emissions.** While MAE and *k* do change together, the MAE in equation 2 is left independent of *k* due to the nature of the two-part emissions parameterization. Equation 1 serves the purpose of determining how absorbing BB OA is, following the parameterization used in Saleh et al. (2014); in their study, this imaginary refractive index is actually referred to as  $k_{OA}$ . We refer to it as  $k_{BrC}$  because, in our study, we
- attribute all organic absorption to BrC. Equation 2 is a separate parameterization from Zhang et al. (2020) in which the absorptivity of OA (expressed as k) is translated into BrC emissions using an assumed BrC MAE (see submitted L171-173). Thus, we use this constant MAE because that

is what's recommended specifically for equation 2 in Zhang et al. If MAE in equation 2 were to be dependent on  $k_{BrC}$ , according to the relation stated in equation 7, the resulting emissions

- 385 proportion would be constant at 100%. It is worth noting again that this parameterization is not expected to give the most accurate emissions ratio, but instead serves as a starting point for sensitivity tests (see submitted L185-186). Regardless, this can be clarified, so we've added the following language to submitted L181: "As all organic absorption is attributed to BrC, equation 2 uses the imaginary RI from equation 1, which indicates the extent of OA absorption, and a BrC
- 390 *MAE value to determine how much BrC emissions would be needed to account for this absorption.*"

- Line 219: What are the values/ranges of parameters used to estimate n and k? We provided these values in the caption of submitted Figure 1 (see submitted L236). However, for clarity, we have also added them to submitted L223: "...(sample AK 4-8" 5% MC from their

- 395 study,  $a=4.554e29 \ s^{-2}$ ,  $\gamma=2.605e13 \ s^{-1}$ ,  $\lambda_0=308 \ nm$ )." We also edited Figure 1 by adding the original datapoints of this sample to demonstrate the good fit of these parameters, editing submitted L237: "These are applied to sample AK 4-8" 5% MC from Sumlin et al. (2018). shown as points along the blue line with data uncertainty displayed in error bars."
- Line 225: How does the 0.003 compare to the default OA imaginary refractive index? This default value is shown under the kOA column of submitted Table 2, but, for clarity, we have added it to submitted L124 where the default OA treatment is first discussed: "...all organics treated as slightly absorbing in the UV-visible wavelength band using an imaginary refractive index (kOA) of 0.00567." Therefore, the weakly absorbing BrC case is slightly lower in magnitude of imaginary refractive index than the current model default which applies to all organics.
  - Line 229: Why is f\_HM = 89%? This value of volume-mixing ratio led to the best fit of the nBrC data for this BrC sample from Sumlin et al. (2018), as discussed in submitted L226-230.
    Line 229: "... led to not only a good fit with n\_BrC spectra..." not clear what is meant here. Please clarify and rephrase if required. Also, what is the range of n BrC using the
- 410 **Kramers-Kronig relations?** We have rephrased this sentence as follows: "...*led to a good fit* with  $n_{BrC}$  spectra and maintained the fit of  $k_{BrC}$  spectra...". The range of  $n_{BrC}$  values used was given by Sumlin et al. (2018). We fit this range by, firstly, using Kramers-Kronig relationships to fit the  $k_{BrC}$  data, and subsequently mixing with a non-absorbing host material to fit  $n_{BrC}$ . The resulting  $n_{BrC}$  spectra ranges from 1.84 at 350 nm to 1.49 at 700 nm. To clarify this, we have
- 415 added these numbers to submitted L230: "Furthermore, taking the solar spectrum weighted average of these  $n_{BrC}$  spectra, which ranged from  $n_{BrC, 350 \text{ nm}} = 1.84$  to  $n_{BrC, 700 \text{ nm}} = 1.49$ , led to a UV-VIS averaged  $n_{BrC} \approx 1.53$ ..."

- Figure 2: This is not very useful in its present form as most of the points are clustered together. It may be revised or represented as a table. We presented the refractive indices in

- 420 this form to show exactly that–SOA imaginary refractive indices are small and clustered together compared to the values of moderately and strongly absorbing primary BrC. However, for the sake of clarity and reproducibility we include the exact values of primary and secondary BrC refractive indices in Table A1, now referenced in submitted L264: "...*imaginary RIs can be seen in Fig. 2, along with RIs for primary BrC and other aerosol tracers, and are listed in Table A1,
  425 for reference."*
  - Line 280: How were the Mie calculations performed? Do the refractive indexes correspond to the k with 20% and 150% of the absorption? The absorption efficiency of emitted BrC was found through Mie calculations using all  $k_{BrC}$  cases (weakly, moderately, and

strongly absorbing), aerosol radius of 0.2 µm (as prescribed in Table 1), and wavelength of 550

- 430 nm. Then, Mie calculations were performed again, using the same wavelength and aerosol radius, but this time iterating through different imaginary refractive indices. RIs that resulted in 150% ("browner") or 20% ("threshold") absorption efficiency relative to each of the emitted cases were used as kBrC for the corresponding aged tracers, as mentioned in submitted L280. We have clarified this with the following edits to submitted L280: "*Iterative Mie calculations were*
- 435 used to determine what imaginary RI, varied from that of emitted BrC, produce the relative absorption efficiencies (150% and 20%) of each aged tracer."
   Line 407-409: How were these biomass burning regions selected? What kind of biomass burning is included here? We added an explanation of how these regions were selected to submitted L407: "When narrowing analysis to peak BB regions and months, we looked at BrC
- 440 emissions hotspots, Southern Hemisphere South America (SHSA), Southern Hemisphere Africa (SHAF), Southeast Asia (SEAS), and Equatorial Asia (EQAS) (Laskin et al., 2015), regions prone to BB and increasingly relevant in recent years, Temperate North America (TENA), Boreal North America (BONA), and Australia (AUST), as well as Boreal Asia (BOAS) to complement analysis of BONA (Fig. 5)." Note, region delineations and acronyms come from Pan
- 445 et al. (2020). Van der Werf et al. (2017) identifies the types of fires that dominate in each of these regions, which are as follows (in decreasing order of magnitude):
  - SEAS-savanna/grassland/shrubland fires, tropical deforestation/degradation, agricultural waste burning (to a lesser extent)
  - SHSA-tropical deforestation/degradation, savanna/grassland/shrubland fires
- EQAS-peat fires, tropical deforestation/degradation
  - SHAF–savanna/grassland/shrubland fires
  - TENA-temperate forest fires, savanna/grassland/shrubland fires, agricultural waste burning (approximately same amount as savanna fires)
  - BONA-boreal forest fires

455

- AUST- savanna/grassland/shrubland fires, temperate forest fires (to a lesser extent)
- BOAS-boreal forest fires, agricultural waste burning (to a lesser extent)

We have added these dominant fire types to submitted Table A2 and have referenced this in submitted L414.

- Line 483: How did Druge et al. (2022) treat BrC-to-OA differently? As mentioned in the
   following line (see submitted L484), Drugé et al. (2022) defined all BB OA as brown, and set all
   fossil fuel OA as non-absorbing. This differs from Brown et al. (2018), which assumed all OA
   are brown, as well as Zhang et al. (2020)/our approach which treated a portion of BB OA as
   brown.
- Figure 9: Why does the only bleaching have a higher radiative effect in comparison to the moderately absorbing (bleaching + browning) case? If this is internal variability, does this mean that bleaching is the key parameter? Is bleaching the dominant process during the BrC lifetime? The radiative effect of the only bleaching case is 0.041 ± 0.01 W m-2, while the radiative effect of the base case (bleaching and browning) is 0.039 ± 0.01 W m-2. The radiative effect of these two simulations is indistinguishable, given they both have a variability of 0.01 W
- 470 m-2. The small difference in magnitude is, as mentioned by this comment, a result of internal model variability. As mentioned in our discussion of submitted Figure 9 (submitted L533-538), when considering a scale of annual global average radiative effect, bleaching is the key parameter. We further emphasize this in Section 5. The chemical lifetimes of emitted and aged BrC are discussed in section 2.2.4 (see submitted L291-301). To briefly summarize, browner

475 BrC (built up overnight) rapidly bleaches during the day. Since BrC can only have a radiative effect when there is insolation, and browner BrC is short lived during daytime, bleaching is the dominant process with regards to radiative effect.

- Line 576: "... total AAOD is usually dominated either by BC or dust aerosols" - what is this statement based on? If this is based on previous studies (reference), do they consider

- 480 **absorption by BrC?** This statement is based on the ModelE relative contributions of dust, BC, and BrC to total AAOD, presented in submitted Figure 12 (manuscript reference on submitted L576) and discussed in more detail in submitted L586. We want to point out the important caveat included in L586-589, which is that BrC may contribute more to total AAOD at shorter wavelengths, which has been previously reported in literature (Laskin et al., 2015).
- 485 Line 593-602: The model overestimates AOD in most cases and underestimates AAOD this is different from what IPCC AR6 models simulate (GliB et al., 2021), which underestimate both the properties. Why might this be happening? If AOD is being dominated by another tracer, would this be overpowering any changes in AOD caused by the introduction of BrC? There seems to be an improvement in simulating AAOD, so why
- 490 **is this not discussed as extensively in the manuscript?** Addressing AOD first: we acknowledged that this overestimation bias is different from previous modeling studies and discussed what could be contributing to this after we demonstrate that the same trend is seen in MODIS data (see submitted L623-636). To briefly summarize, this AOD overestimation could be coming from a change in the ModelE radiation scheme unrelated to this current work; there
- 495 was a change made in optical calculation and the ModelE natural emissions haven't yet been retuned (submitted L625-627 has been updated to state this more clearly: "...but the ModelE radiation scheme has since been updated with a change in optical calculation, including a more accurate treatment of aerosol hydration using Köhler theory. The ModelE natural emissions haven't yet been retuned following this change.") Over the SEAS region, the bias could also be
- due to a possible overestimation of BB emissions. Regarding the idea of total AOD overpowering any changes caused by BrC introduction, we have added the following text to submitted L637 to address this: "We do not expect these biases to overpower BrC-driven changes in total AOD, because the BrC contribution to total AOD is small-approximately 5% of average total AOD in the base case-and, as previously stated, there was no distinguishable
  change in total AOD across all sensitivity test simulations, including the control case."

Regarding AAOD: As stated in submitted L600-601, we don't draw any substantive conclusions from this because of the limited number of sites with data. Further, we disagree that the right column of submitted Figure 11 shows an improvement in stimulating AAOD; though the slope

- of the linear regression between AERONET and ModelE moves closer to one, the correlation coefficient is still small, again limiting our ability to draw a conclusion.
  Line 601: "... data scatter is mostly caused by dust and BC, rather than BrC" how was this determined? We concluded this because dust and BC dominate the AAOD signal. Higher magnitude AAODs would be expected to dominate the scatter. Submitted Figure 10 shows BrC
- 515 variability in AAOD is on the order of 10-6-10-7, confirming it contributes indistinguishably to total AAOD scatter. To clarify this, we've drawn attention back to submitted Fig. 10 at the end of submitted L602: "...data scatter is mostly caused by dust and BC, rather than BrC. which shows minimal variability in AAOD (see Fig. 10)."

---

## Author Response (AR2)

We thank the reviewer for the helpful comments and suggested minor revisions that have improved the manuscript. This document includes comments from the reviewer as well as our responses. Reviewer comments are **bolded**, responses are in regular font, and *excerpts of changes to manuscript are in italics, with new changes underlined if added to an existing sentence.* Any new references cited in manuscript excerpts have been added to the references section of the manuscript.

**The authors have addressed the comments raised by the reviewers and the authors must be commended for developing an improved methodology to represent BrC absorption in climate models. A few minor edits are suggested below; I recommend publication of the manuscript after they have been addressed.**

**1) Equations 1 and 2 indicate the imaginary refractive index as k_BrC as opposed to k_OA. This is misleading as the imaginary refractive index calculated by Saleh et al. (2014) is for the OA mass and not the BrC mass. Also, equation 2 only holds if it is k_OA!**
**E_BrC = (4.π.k_OA/ρ.λ)/MAC_BrC × E_OA (Equation 2)**
**E_BrC × MAC_BrC = MAC_OA × E_OA (or the absorption from BrC = absorption from OA).**

We specify $k_{BrC}$ in equations 1 and 2, rather than $k_{OA}$, because we treat all absorption from OA as BrC absorption (as suggested by the reviewer). This treatment wasn't explicitly stated in the manuscript however, which led to the misleading nature of these equations. To resolve this, we have added the following text to revised submission L150: *"In this scheme, we consider total OA to consist of non-absorbing OA and BrC; any organic absorption is attributed to BrC."* We also added clarifying text to revised submission L175 to acknowledge the difference in $k$ specified by Saleh et al. and our study: *"Since all organic absorption in our scheme is attributed to BrC, we use $k_{BrC}$ in place of the $k_{OA}$ specified by Saleh et al. (2014) for equation 1."*

**2) The study limitations (section 3.4) include possible missing absorbing species and a limited understanding of the browning and bleaching of BrC. Differences between the optical properties of water-soluble and insoluble fractions may also be included here (Zhang et al., 2020; Liu et al., 2013; Satish et al., 2020; Laskin et al., 2015; Saleh et al., 2020). While the study does a great job of using presently available information on BrC absorption properties, further improvements on these limitations may change the findings of the limited influence of BrC refractive index or browning when considering the global radiative effect of BrC. For example, if water-insoluble darker BrC are included and have a reduced susceptibility to photo-bleaching (of course, only after experimental data becomes available), then their effect may be pronounced even at a global scale. This possibility may be acknowledged in the manuscript.**

We agree this limitation should be discussed. We added the following paragraph to section 3.4, starting on revised submission L687: *"Another limitation of this scheme is that water-soluble and water-insoluble fractions of BrC were not differentiated or characterized with different optical properties and aging. This was done to be consistent with pre-existing OA representation in ModelE, as BB OA and biogenic SOA are not differentiated by solubility or hygroscopicity (Koch, 2001), and as mentioned in section 2.2.3, changes to prescribed BrC solubility affect total organic burden and skew estimates BrC radiative effect. Studies have shown, however, that water-insoluble BrC can be more absorbing than water-soluble BrC (Chen and Bond, 2010; Liu*

*et al., 2013; Laskin et al., 2015; Satish et al., 2020). It is possible that chemical aging also differs between these two BrC types, for instance darker water-insoluble BrC being more resistant to bleaching, since reactions may proceed faster in aqueous phase (Hems et al., 2021).*
50 *Accounting for these differences in BrC solubility types could change model sensitivity to refractive index and aging. Further study on aging in water-soluble and water-insoluble conditions could clarify the potential impact of not differentiating BrC by solubility within ModelE."*

55 **3) In the response document (Line 271) it is shown that fixed-time ageing simulations show little variation from the base case that uses an oxidant-concentration-driven ageing scheme in terms of global RF but an oxidant-concentration-driven aging scheme has better spatial accuracies. Is this also not true for other properties like the BrC refractive index?**
In the response document, we identified spatial differences that may be important as they overlap
60 with some biomass burning regions. We can't conclude, however, that these differences are necessarily more accurate. The oxidant-driven aging scheme was characterized as more accurate in the sense that it is more physically accurate because we know BrC aging changes with oxidant concentration–we differ to the scheme that's more reflective of observed chemistry. With regards to the reviewer question about refractive index: the effective refractive of BrC, an evolving value
65 observed in the atmosphere, is represented in ModelE as a mix of multiple BrC tracers. Since BrC absorbs more with greater imaginary RI, we would expect similar spatial differences between fixed-aging and oxidant-driven chemistry for refractive index as we did with radiative effect. Without more observations of BrC RI however, which the recently launched PACE mission will afford us, we can't conclude if these differences are more accurate.
70
**4) From the response document, Line 471: "To briefly summarize, browner BrC (built up overnight) rapidly bleaches during the day. Since BrC can only have a radiative effect when there is insolation, and browner BrC is short-lived during daytime, bleaching is the dominant process with regards to radiative effect.". This seems important in understanding**
75 **why browning may not have a significant radiative effect and may be mentioned in the manuscript if it hasn't been mentioned explicitly already.**
A sentence stating this explicitly has been added to revised submission L663, following the statement that BrC radiative effect shows limited change with and without browning: *"Such limited sensitivity to browning makes sense: BrC can only have a radiative effect when there is*
80 *insolation, and browner BrC is short-lived during daytime, so bleaching is the dominant process with regards to radiative effect."*

**References:**
**Zhang et al., 2020: doi:10.5194/acp-20-4889-2020**
85 **Liu et al., 2013: doi:10.5194/acp-13-12389-2013**
**Satish et al., 2020: doi:10.1007/s11356-020-09388-7**